# *Drosophila* models of pathogenic copy-number variant genes show global and non-neuronal defects during development

Tanzeen Yusuff[1], Matthew Jensen[1], Sneha Yennawar[1], Lucilla Pizzo[1], Siddharth Karthikeyan[1], Dagny J. Gould[1], Avik Sarker[1], Erika Gedvilaite[1], Yurika Matsui[1,2], Janani Iyer[1], Zhi-Chun Lai[1,2], Santhosh Girirajan[1,3] *

1 Department of Biochemistry and Molecular Biology, Pennsylvania State University, University Park, Pennsylvania, United States of America, 2 Department of Biology, Pennsylvania State University, University Park, Pennsylvania, United States of America, 3 Department of Anthropology, Pennsylvania State University, University Park, Pennsylvania, United States of America

☯ These authors contributed equally to this work.
* sxg47@psu.edu

**Data Availability Statement:** All relevant data are within the manuscript and its Supporting Information files.

## Abstract

While rare pathogenic copy-number variants (CNVs) are associated with both neuronal and non-neuronal phenotypes, functional studies evaluating these regions have focused on the molecular basis of neuronal defects. We report a systematic functional analysis of non-neuronal defects for homologs of 59 genes within ten pathogenic CNVs and 20 neurodevelopmental genes in *Drosophila melanogaster*. Using wing-specific knockdown of 136 RNA interference lines, we identified qualitative and quantitative phenotypes in 72/79 homologs, including 21 lines with severe wing defects and six lines with lethality. In fact, we found that 10/31 homologs of CNV genes also showed complete or partial lethality at larval or pupal stages with ubiquitous knockdown. Comparisons between eye and wing-specific knockdown of 37/45 homologs showed both neuronal and non-neuronal defects, but with no correlation in the severity of defects. We further observed disruptions in cell proliferation and apoptosis in larval wing discs for 23/27 homologs, and altered Wnt, Hedgehog and Notch signaling for 9/14 homologs, including *AATF/Aatf*, *PPP4C/Pp4-19C*, and *KIF11/Klp61F*. These findings were further supported by tissue-specific differences in expression patterns of human CNV genes, as well as connectivity of CNV genes to signaling pathway genes in brain, heart and kidney-specific networks. Our findings suggest that multiple genes within each CNV differentially affect both global and tissue-specific developmental processes within conserved pathways, and that their roles are not restricted to neuronal functions.

## Author summary

Rare copy-number variants (CNVs), or large deletions and duplications in the genome, are associated with both neuronal and non-neuronal clinical features. Previous functional studies for these disorders have primarily focused on understanding the cellular mechanisms for neurological and behavioral phenotypes. To understand how genes within these

**Funding:** This work was supported by National Institute of General Medical Sciences (NIGMS; https://www.nigms.nih.gov/) R01-GM121907 and resources from the Huck Institutes of the Life Sciences (https://huck.psu.edu) to S.G., and NIGMS T32-GM102057 to M.J. The funders had no role in study design, data collection and analysis, decision to publish, or preparation of the manuscript.

**Competing interests:** The authors have declared that no competing interests exist.

CNVs contribute to developmental defects in non-neuronal tissues, we assessed 79 homologs of CNV and known neurodevelopmental genes in *Drosophila* models. We found that most homologs showed developmental defects when knocked down in the adult fly wing, ranging from mild size changes to severe wrinkled wings or lethality. Although a majority of tested homologs showed defects when knocked down specifically in wings or eyes, we found no correlation in the severity of the observed defects in these two tissues. A subset of the homologs showed disruptions in cellular processes in the developing fly wing, including alterations in cell proliferation, apoptosis, and cellular signaling pathways. Furthermore, human CNV genes also showed differences in gene expression patterns and interactions with signaling pathway genes across multiple human tissues. Our findings suggest that genes within CNV disorders affect global developmental processes in both neuronal and non-neuronal tissues.

## Introduction

Rare copy-number variants (CNVs), or deletions and duplications in the genome, are associated with neurodevelopmental disorders such as autism, intellectual disability (ID), and schizophrenia [1,2]. While dosage alteration of CNV regions contribute predominantly to defects in nervous system development, several CNV-associated disorders also lead to early developmental features involving other organ systems [3,4], including cardiac defects [5,6], kidney malformations [7], craniofacial features [3], and skeletal abnormalities [8]. In fact, an overall survey of ten rare disease-associated CNVs among individuals within the DECIPHER database [9] showed a wide range of non-neuronal phenotypes across multiple organ systems for each CNV disorder (S1 Fig, S1 Data). For example, the 1q21.1 deletion causes variable expression of multiple neuronal and non-neuronal phenotypes, including developmental delay, autism, and schizophrenia as well as craniofacial features, cataracts, cardiac defects, and skeletal abnormalities [10–12]. Additionally, while the 7q11.23 deletion associated with Williams-Beuren syndrome (WBS) causes neuropsychiatric and behavioral features, other non-neuronal phenotypes, including supravalvular aortic stenosis, auditory defects, hypertension, diabetes mellitus, and musculoskeletal and connective tissue anomalies, are also observed among the deletion carriers [13].

Despite the prevalence of non-neuronal phenotypes among CNV carriers, functional studies of CNV genes have primarily focused on detailed assessments of neuronal and behavioral phenotypes in model systems. For example, mouse models for the 16p11.2 deletion exhibited post-natal lethality, reduced brain size and neural progenitor cell count, motor and habituation defects, synaptic defects, and behavioral defects [14–16]. Similarly, mouse models for the 3q29 deletion showed decreased weight and brain size, increased locomotor activity, increased startle response, and decreased spatial learning and memory [17,18]. However, fewer studies have focused on detailed evaluation of non-neuronal phenotypes in functional models of CNV disorders. For example, Arbogast and colleagues evaluated obesity and metabolic changes in 16p11.2 deletion mice, which showed reduced weight and impaired adipogenesis [19]. While Haller and colleagues showed that mice with knockdown of *MAZ*, a gene within the 16p11.2 deletion region, exhibit genitourinary defects also observed in individuals with the deletion [20], mouse studies for other homologs of 16p11.2 genes, including *TAOK2*, *KCTD13*, and *MAPK3*, have only focused on assessing neuronal defects [21–25]. Furthermore, Dickinson and colleagues reported a high-throughput analysis of essential genes in mice and identified both neuronal and non-neuronal phenotypes for individual gene knockouts, including more

than 400 genes that lead to lethality [26]. While these efforts aided in implicating novel genes with human disease, our understanding of how genes associated with neurodevelopmental disorders contribute towards non-neuronal phenotypes is still limited. Therefore, a large-scale analysis of non-neuronal phenotypes is necessary to identify specific candidate genes within CNV regions and associated biological mechanisms that contribute towards these phenotypes.

*Drosophila melanogaster* is an excellent model system to evaluate homologs of neurodevelopmental genes, as many developmental processes and signaling pathways are conserved between humans and flies [27]. In fact, over 75% of human disease genes have homologs in *Drosophila*, including many genes involved in cellular signaling processes [28,29]. We recently examined the contributions of individual *Drosophila* homologs of 28 genes within the 16p11.2 and 3q29 deletion regions towards specific neurodevelopmental phenotypes, including rough eye phenotypes and defects in climbing ability, axon targeting, neuromuscular junction, and dendritic arborization [30,31]. While these findings implicated multiple genes within each CNV region towards conserved cellular processes in neuronal tissues, the conserved role of these genes in non-neuronal tissues is not well understood. The *Drosophila* wing is an effective model system to evaluate such developmental defects, as key components of conserved signaling pathways, such as Notch, epidermal growth factor receptor (EGFR), Hegdehog, and Wnt pathways, were identified using fly wing models [32–38]. Although fly wing phenotypes cannot be directly translated to human phenotypes, defects observed in fly wings can be used to assess how homologs of human disease genes alter conserved cellular and developmental mechanisms. For example, Wu and colleagues showed that overexpression of the *Drosophila* homolog for *UBE3A*, associated with Angelman syndrome, leads to abnormal wing and eye morphology defects [39]. Furthermore, *Drosophila* mutant screens for developmental phenotypes, including wing defects, were used to identify conserved genes for several human genetic diseases, including Charcot-Marie-Tooth disease and syndromic microcephaly [40]. Kochinke and colleagues also recently performed a large-scale screening of ID-associated genes, and found an enrichment of wing trichome density and missing vein phenotypes in ID genes compared to control gene sets [41]. Hence, the fly wing provides a model system that is ideal for evaluating the contributions of individual homologs of CNV genes towards cellular and developmental defects.

In this study, we tested tissue-specific and cellular phenotypes of 79 total fly homologs, including 59 fly homologs of human genes within ten pathogenic CNV regions and 20 genes associated with neurodevelopmental disorders. We used the adult fly wing to evaluate phenotypes in a non-neuronal tissue, and observed a wide range of robust qualitative and quantitative wing phenotypes among 136 RNA interference (RNAi) lines, including size defects, ectopic and missing veins, severe wrinkling, and lethality. Further analysis of cellular phenotypes revealed disruptions in conserved developmental processes in the larval imaginal wing disc, including altered levels of cell proliferation and apoptosis as well as altered expression patterns in the Wnt, Hedgehog, and Notch signaling pathways. However, we found no correlation in the severity of phenotypes observed with wing and eye-specific knockdown. These findings were further supported by differences in expression patterns and network connectivity of human CNV genes across different tissues. Our analysis emphasizes the importance of multiple genes within each CNV region towards both global and tissue-specific developmental processes.

## Results

### Knockdown of fly homologs of CNV genes contribute to a range of wing defects

We used the $bx^{MS1096}$-GAL4 wing-specific driver to assess a total of 136 RNAi lines for 59 fly homologs of 130 total human genes within pathogenic CNV regions (chromosomal locations

1q21.1, 3q29, 7q11.23, 15q11.2, 15q13.3, 16p11.2, distal 16p11.2, 16p12.1, 16p13.11, and 17q12), as well as 20 fly homologs of human genes associated with neurodevelopmental disorders (S2 Data). Fly homologs of these genes were identified using the DIOPT orthology prediction tool (S2 Data) [42]. We list both the human gene name and the fly gene name for each tested gene as *HUMAN GENE/Fly gene* (i.e. *KCTD13/CG10465*), as well as the human CNV region for context at first instance. We scored 20–25 adult wings for five distinct wing phenotypes in each non-lethal RNAi line, including wrinkled wing, discoloration, ectopic veins, missing veins, and bristle planar polarity phenotypes (Fig 1; S3 Data). We categorized adult female wing phenotypes based on their severity, and performed k-means clustering analysis to categorize each RNAi line by their overall phenotypic severity (Fig 2A and 2B). We observed four clusters of RNAi lines: 75 lines with no observable qualitative phenotypes (55.2%), 24 lines with mild phenotypes (17.6%), 10 lines with moderate phenotypes (7.4%), 21 lines with severe phenotypes (15.4%), and 6 lines with lethal phenotypes (4.4%), including *ACACA/ACC* within 17q12, *DLG1/dlg1* within 3q29, and *STX1A/Syx1A* within 7q11.23 (Fig 2; S3 Data). We observed severe wrinkled wing phenotypes for 13/79 fly homologs, including *PPP4C/Pp4-19C* within 16p11.2, *ATXN2L/Atx2* within distal 16p11.2, *AATF/Aatf* within 17q12, and *MFI2/Tsf2* within 3q29 (Fig 3A and 3B, S4 Data). Interestingly, seven out of ten CNV regions contained at least one homolog that showed lethality or severe wing phenotype, and five CNV regions (3q29, 16p11.2, distal 16p11.2, 16p12.1, and 17q12) had multiple homologs showing lethality or severe wing phenotypes (Fig 3A, S4 Data). For example, RNAi lines for both *UQCRC2/UQCR-C2* and *POLR3E/Sin* within 16p12.1 showed lethality. Within the 3q29 region, *NCBP2/Cbp20* and *MFI2/Tsf2* showed severe phenotypes, while *DLG1/dlg1* showed lethality. In contrast, 12/20 known neurodevelopmental genes showed no observable wing phenotypes, suggesting that these genes could be responsible for neuronal-specific phenotypes (Fig 3B, S4 Data). We note that 18/79 fly homologs showed discordant phenotypes between two or more RNAi lines for the same gene, which could be due to differences in expression of the RNAi construct among these lines (S4 Data).

Certain qualitative phenotypes were present at a higher frequency in males compared to females among all non-lethal and non-severe RNAi lines (independent of k-means clustering analysis). For example, discoloration (87 lines in males compared to 55 lines in females; $p = 1.315 \times 10^{-4}$, two-tailed Fisher's exact test) and missing vein phenotypes (92 lines in males compared to 29 lines in females; $p = 2.848 \times 10^{-16}$, two-tailed Fisher's exact test) were more commonly observed in males than females (S3 Data). In particular, 25/92 lines in males (compared to 1/29 lines in females) showed a total loss of the anterior crossvein (ACV) (S3 Data). We further identified 17 RNAi lines that were lethal in males with wing-specific knockdown of fly homologs. While higher frequencies of wing phenotypes in males could be due to a sex-specific bias of developmental phenotypes, the increased severity we observed in males is most likely due to a stronger RNAi knockdown caused by X-linked dosage compensation, as the $bx^{MS1096}$-GAL4 driver is inserted on the fly X chromosome [43,44].

Next, we measured the total adult wing area and lengths of six veins (longitudinal L2, L3, L4, L5, ACV, and posterior crossvein or PCV) in the adult wing for each of the tested RNAi lines that did not show lethality (or severe wrinkled phenotypes for vein length measurements) (Fig 4A). Overall, we identified significant wing measurement changes for 89 RNAi lines compared to controls, which included lines that did not have an observable qualitative wing phenotype (Fig 2C, S3 Data). A summary of L3 vein lengths is presented in Fig 4B, and the measurements for the remaining five veins are provided in S2 Fig and S3 Data. We found that 33/61 homologs (54.1%) showed significant changes in L3 vein length, including 20 homologs with longer vein lengths and 13 homologs with shorter vein lengths (S4 Data). Additionally, 41/74 fly homologs (55.4%) showed changes in wing area, including 36 homologs which

## Experimental strategy

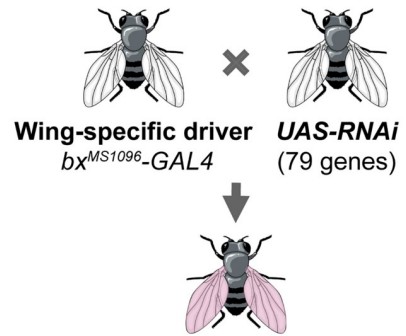

**Wing-specific driver** *bx^{MS1096}-GAL4* ✕ *UAS-RNAi* (79 genes)

**Phenotyping of adult wings**

| **Qualitative analysis** | **Quantitative analysis** |
|---|---|
| • Wrinkled wings | • Longitudinal veins (L2, L3, L4, L5) |
| • Ectopic veins | • Crossveins (ACV, PCV) |
| • Missing veins | • Wing area |
| • Discoloration | |
| • Bristle planar polarity | |

**Tissue-specific effects**

- Compare to ubiquitous knockdown
- Compare to eye-specific knockdown
- Gene expression across multiple tissues (human and flies)

**Cellular processes and developmental pathways**

- Cell proliferation
- Apoptosis
- Disruptions in signaling pathways
- Human tissue-specific network analysis (brain, heart, kidney)

**Fig 1. Experimental approach to identify developmental phenotypes of homologs of CNV genes in the *Drosophila* wing.** Strategy for identifying non-neuronal phenotypes and underlying cellular mechanisms for homologs of CNV and known neurodevelopmental genes using the fly wing as a model system. We evaluated 59 *Drosophila* homologs of genes within 10 CNV regions and 20 known neurodevelopmental genes (79 total homologs). Using the *UAS-GAL4* system with wing-specific *bx^{MS1096}* driver, we knocked down 136 individual RNAi lines for homologs of CNV and neurodevelopmental genes, and evaluated qualitative and quantitative phenotypes. We next clustered RNAi lines based on severity of qualitative phenotypes, and compared adult wing phenotypes to phenotypes observed with ubiquitous and eye-specific knockdown of homologs. Furthermore, we evaluated underlying cellular mechanisms for the observed wing-specific phenotypes, and examined the connectivity patterns of candidate homologs for developmental phenotypes in multiple human tissue-specific gene networks.

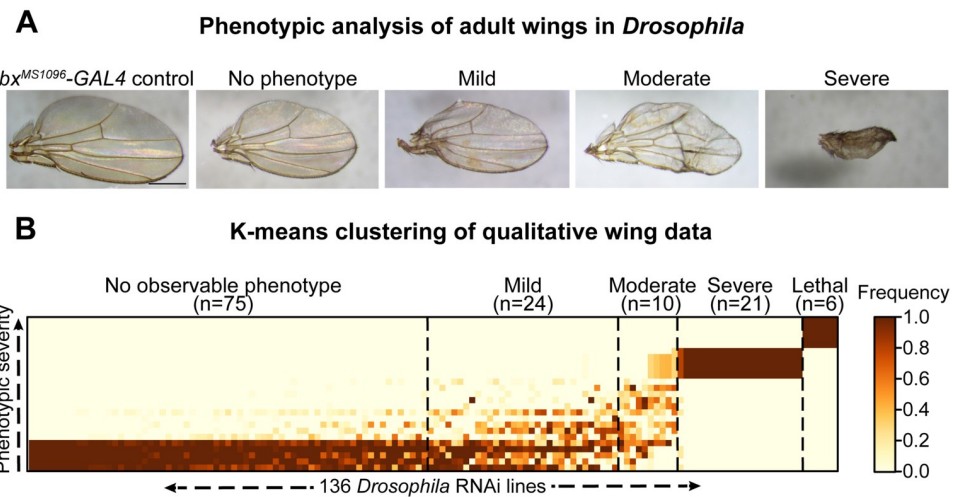

**Fig 2. Global analysis of developmental phenotypes with knockdown of homologs of CNV and neurodevelopmental genes.** (A) Representative brightfield images of adult wing phenotype severity observed with knockdown of homologs of CNV and neurodevelopmental genes, based on clustering analysis, are shown (scale bar = 500μm). (B) Heatmap with k-means clustering of qualitative phenotypes in adult female wings across 136 RNAi lines is shown. The color of each cell represents the frequency of individual fly wings (n = 20–25 adult wings) for each RNAi line (x-axis) that show a specific severity (no phenotype, mild, moderate, severe, lethal) for the five qualitative phenotypes assessed (y-axis; wrinkled wings, ectopic veins, missing veins, discoloration, bristle planar polarity), as detailed in S3 Data. Based on these data, we identified clusters for no phenotype (n = 75 lines), mild (n = 24 lines), moderate (n = 10 lines), severe (n = 21 lines), and lethal (n = 6 lines) RNAi lines. (C) Summary table for qualitative and quantitative adult wing phenotypes of all tested RNAi lines for homologs of CNV and neurodevelopmental genes. Quantitative phenotype totals do not include lethal RNAi lines for both area and vein length. In addition, L3 vein length totals do not include RNAi lines with severe phenotypes.

showed smaller wing areas and five homologs showed larger wing areas compared to controls (S4 Data). For example, both homologs of genes within 1q21.1 region, *BCL9/lgs* and *FMO5/Fmo-2*, showed decreased wing area and vein length, potentially mirroring the reduced body length phenotype observed in mouse models of the deletion [45] (Fig 4B and 4C). In addition, *PAK2/Pak* within 3q29, *TBX1/org-1* within 22q11.2, autism-associated *CHD8/kis*, and microcephaly-associated *ASPM/asp* also showed smaller wing areas and vein lengths (Fig 4B and 4C). In contrast, *TRPM1/Trpm* within 15q13.3 and the cell proliferation gene *PTEN/Pten* [46] both showed larger wing areas and vein lengths (Fig 4B and 4C). Furthermore, we identified eight homologs that showed no qualitative wing phenotypes, but had significant changes in wing areas and vein lengths, including *CCDC101/Sgf29* in distal 16p11.2, *FMO5/Fmo-2*, *TRPM1/Trpm*, *DHRS11/CG9150* in 17q12, and *NSUN5/Nsun5* in 7q11.23 (Fig 4B and 4C; S4 Data). These results indicate that homologs of certain CNV genes may influence variations in size without causing adverse wing phenotypes, and may be specifically implicated towards cellular growth mechanisms.

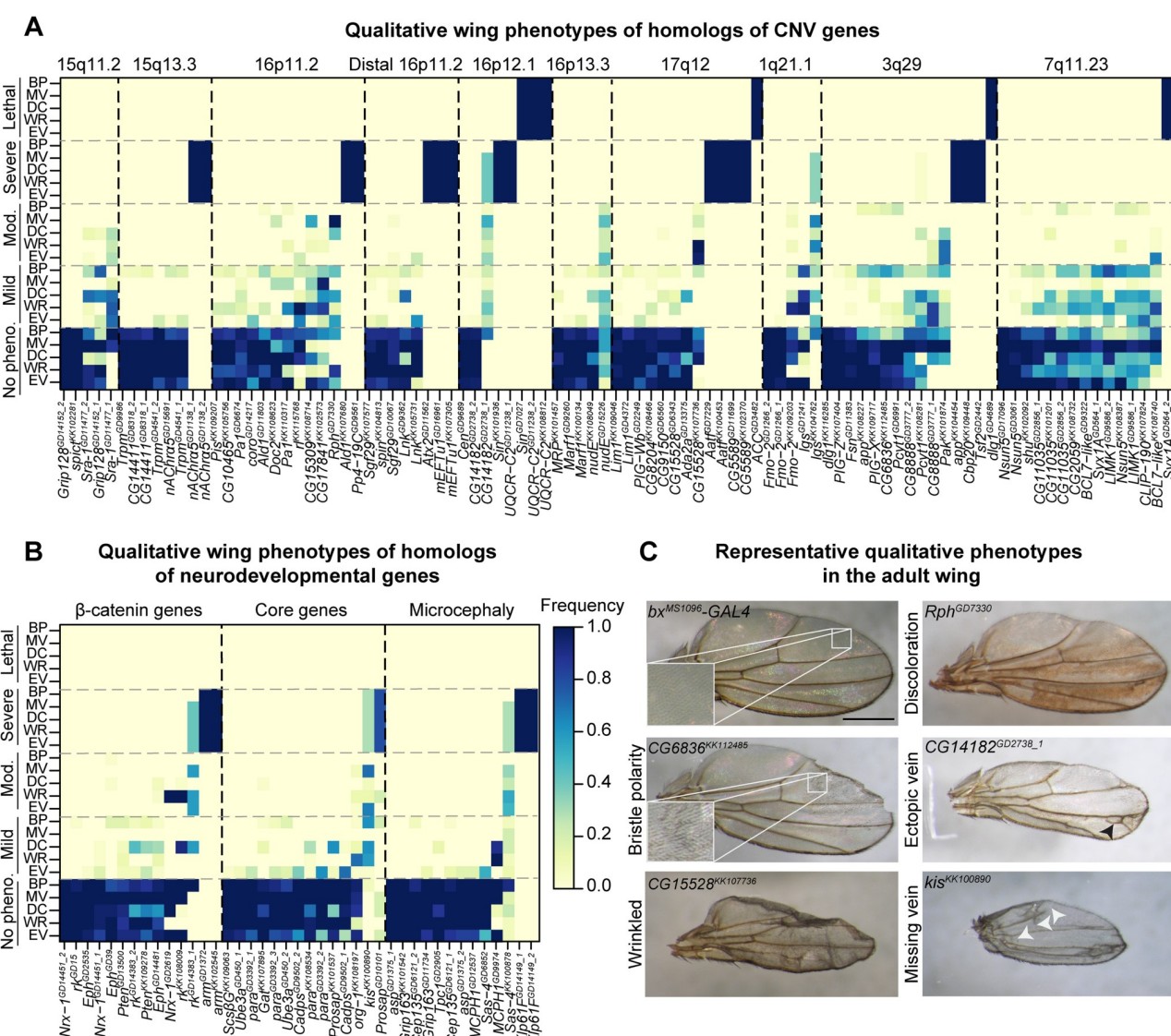

**Fig 3. Qualitative adult wing phenotypes of *Drosophila* homologs of CNV and neurodevelopmental genes.** Heatmaps representing five qualitative adult wing phenotypes for all 136 RNAi lines, with (**A**) 59 tested homologs for 10 CNV regions and (**B**) 20 homologs for neurodevelopmental genes (β-catenin, core neurodevelopmental genes, and microcephaly genes), are shown. The color of each cell represents the frequency of each qualitative phenotype by severity (wrinkled wings, WR; ectopic veins, EV; missing veins, MV; discoloration, DC; bristle planar polarity, BP), ranging from no phenotype to lethal. (**C**) Representative brightfield images of adult fly wings (scale bar = 500μm) with wing-specific knockdown of homologs of CNV and neurodevelopmental genes show the five assessed qualitative phenotypes. Panels in the $bx^{MS1096}$-GAL4 control and $CG6836^{KK112485}$ images highlight bristle planar polarity phenotypes for the representative images. Black arrowheads highlight ectopic veins, and white arrowheads highlight missing veins. Genotypes for the images are: $w^{1118}/bx^{MS1096}$-GAL4;+; UAS-Dicer2/+, $w^{1118}/bx^{MS1096}$-GAL4;UAS-$Rph^{GD7330}$ RNAi/+;UAS-Dicer2/+, $w^{1118}/bx^{MS1096}$-GAL4;UAS-$CG15528^{KK107736}$ RNAi/+; UAS-Dicer2/+, $w^{1118}/bx^{MS1096}$-GAL4;UAS-$CG6836^{KK112485}$ RNAi/+; UAS-Dicer2/+, $w^{1118}/bx^{MS1096}$-GAL4;+;UAS-$CG14182^{GD2738}$ RNAi/UAS-Dicer2, and $w^{1118}/bx^{MS1096}$-GAL4;UAS-$kis^{KK100890}$ RNAi/+; UAS-Dicer2/+.

## Homologs of CNV genes show global and tissue-specific effects during development

We previously showed that many fly homologs of CNV genes that showed wing defects in the current study also contributed towards neuronal phenotypes in the fly eye [30,31], suggesting a role for these genes in global development. We therefore performed ubiquitous and eye-

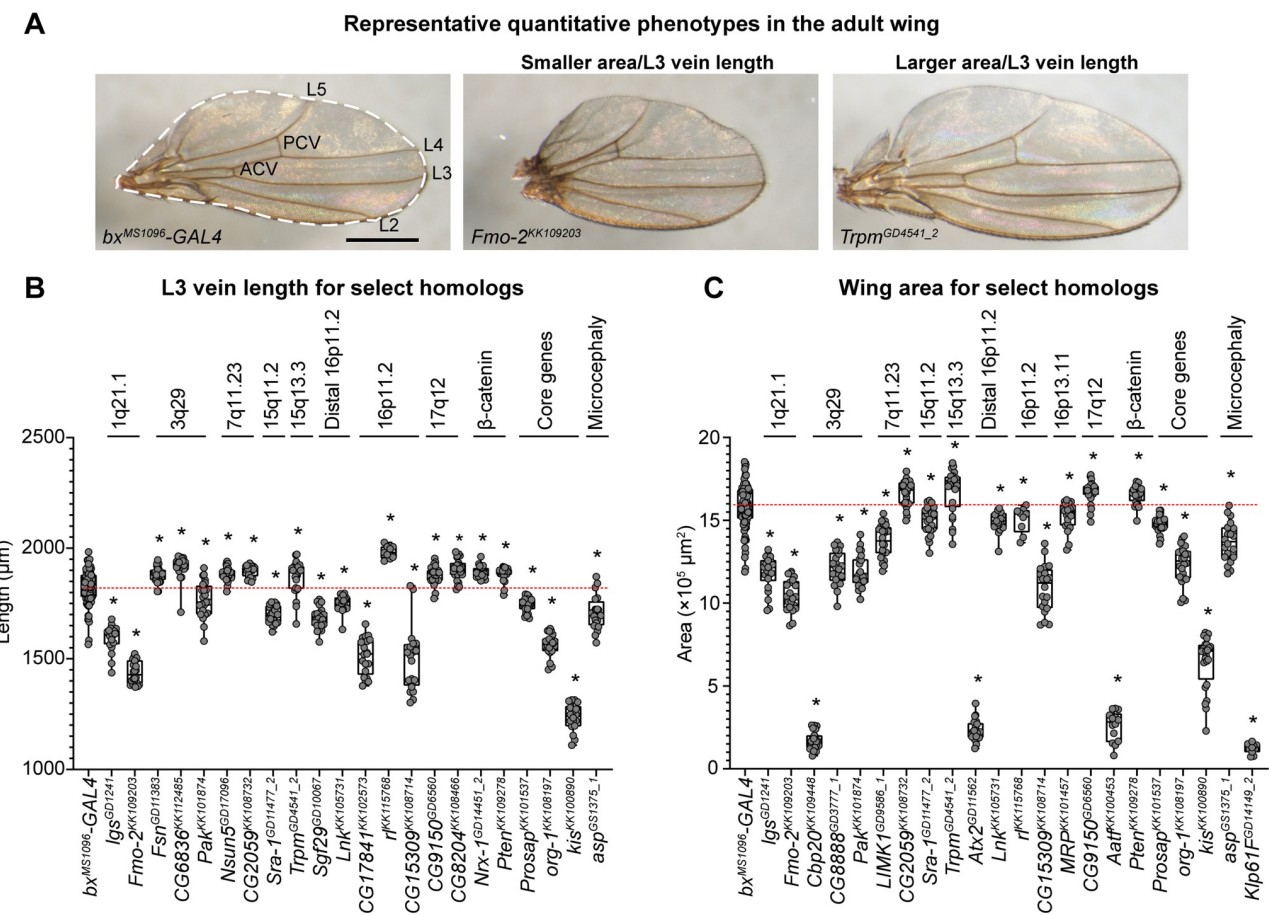

**Fig 4. Quantitative adult wing phenotypes of *Drosophila* homologs of CNV and neurodevelopmental genes. (A)** Representative brightfield images of adult fly wings (scale bar = 500μm) with wing-specific knockdown of homologs of CNV and neurodevelopmental genes with size defects are shown. The $bx^{MS1096}$-GAL4 control image highlights six veins, including longitudinal veins L2, L3, L4, and L5 as well as the anterior and posterior crossveins (ACV and PCV), that were measured for quantitative analysis. The dotted line in the control image represents the total wing area calculated for each RNAi line. Genotypes for the images are: $w^{1118}/bx^{MS1096}$-GAL4;+; UAS-Dicer2/+, $w^{1118}/bx^{MS1096}$-GAL4;UAS-Fmo-$2^{KK109203}$ RNAi/+; UAS-Dicer2/+, and $w^{1118}/bx^{MS1096}$-GAL4;+;UAS-Trpm$^{GD4541}$ RNAi/UAS-Dicer2. **(B)** Boxplot of L3 vein lengths for knockdown of select homologs in adult fly wings (n = 9–91, *p < 0.05, two-tailed Mann–Whitney test with Benjamini-Hochberg correction) is shown. Vein measurements for all other longitudinal veins and crossveins for these lines are represented in S3 Fig. **(C)** Boxplot of wing areas for knockdown of select homologs in adult fly wings (n = 9–91, *p < 0.05, two-tailed Mann–Whitney test with Benjamini-Hochberg correction) is shown. Boxplots indicate median (center line), 25th and 75th percentiles (bounds of box), and minimum and maximum (whiskers), with red dotted lines representing the control median.

specific knockdown of fly homologs to assess tissue-specific effects in comparison to the wing phenotypes. First, we used the *da-GAL4* driver at 25˚C to drive ubiquitous knockdown of RNAi lines for 31 homologs of CNV genes, including 19 that were previously published [30,31]. We observed complete or partial lethality at larval and pupal stages with knockdown of 10/31 homologs (32.3%) (Fig 5A). Lethal phenotypes have also been documented for 43/130 knockout mouse models of individual CNV genes as well as for entire deletion regions (S5 Data). For example, mouse models heterozygous for the 16p11.2 deletion showed partial neonatal lethality, while knockout mouse models of four individual homologs of genes within the 16p11.2 region, including $Ppp4C^{-/-}$ and $Kif22^{-/-}$, showed embryonic lethality [14,47,48]. In our study, the *DLG1/dlg1* line that showed lethality with wing-specific knockdown also exhibited larval lethality with ubiquitous knockdown, indicating its role in global development (Fig 5A). In addition, six homologs that showed severe wing phenotypes also showed larval or pupal

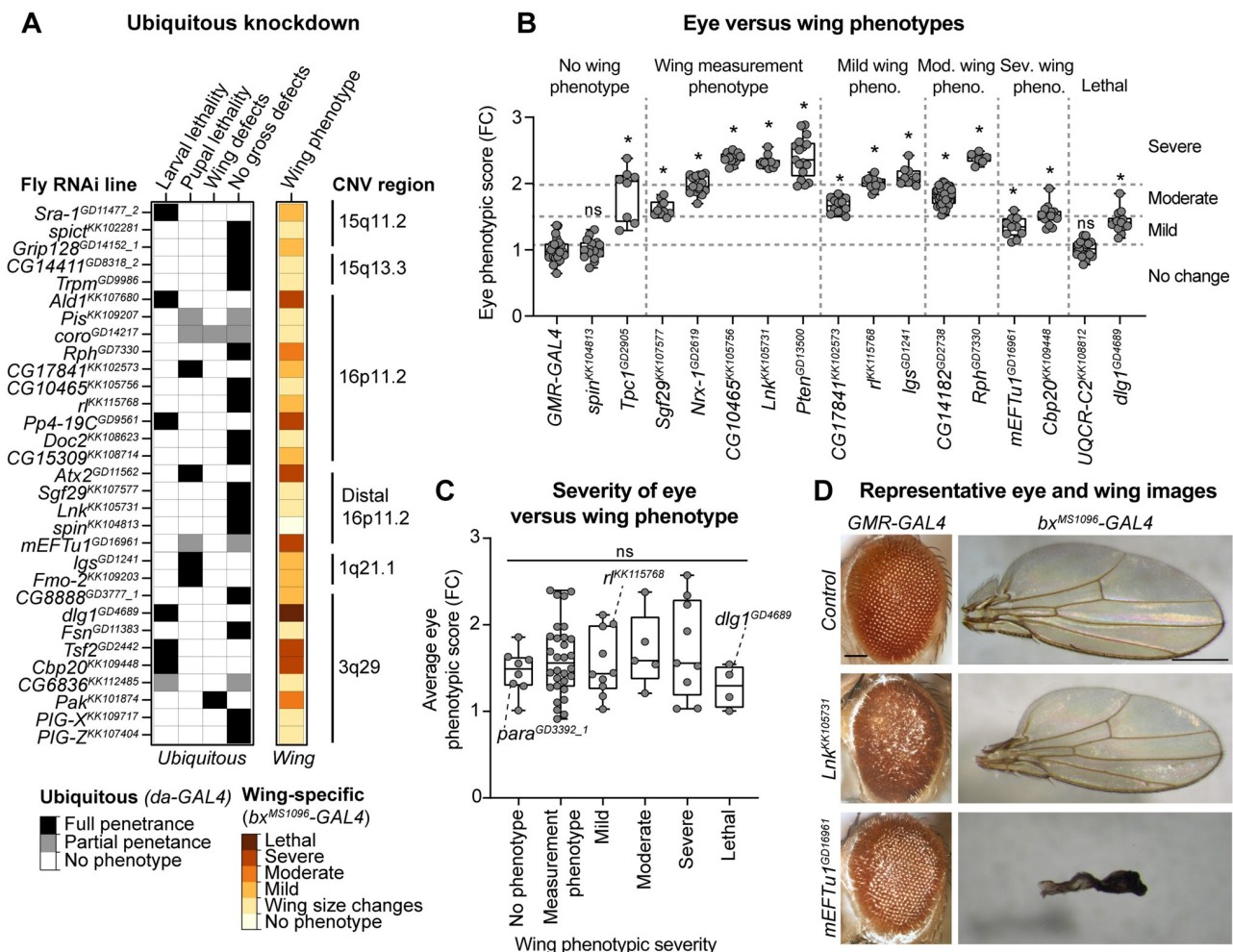

**Fig 5. Comparison of wing-specific, eye-specific, and ubiquitous knockdown of fly homologs.** (A) Heatmap with the penetrance of phenotypes with ubiquitous knockdown (*da–GAL4*) of select homologs of CNV genes, compared to their adult wing-specific (*bx^MS1096^–GAL4*) phenotypic severity, is shown. (B) Boxplots of *Flynotyper*-derived phenotypic scores for adult eyes with eye-specific knockdown (*GMR-GAL4*) of select fly homologs, normalized as fold-change (FC) to control values, are shown (n = 7–40, *p < 0.05, one-tailed Mann–Whitney test with Benjamini-Hochberg correction). Boxplots are arranged by severity of adult wing phenotypes, while eye phenotypic scores are categorized by severity: no change (0–1.1 FC), mild (1.1–1.5 FC), moderate (1.5–2.0 FC), and severe (>2.0 FC). (C) Boxplot shows average eye phenotypic scores for 66 RNAi lines of select fly homologs, normalized as FC to control values, according to wing phenotypic category (n = 4–30 RNAi lines per group). We did not observe significant changes in eye phenotype severity across the five wing phenotypic categories (Kruskal-Wallis rank sum test, p = 0.567, df = 5, $\chi^2$ = 3.881). Examples of average eye phenotypic scores for RNAi lines with no phenotype (*para^GD3392_1^*), mild (*rl^KK115768^*), and lethal (*dlg1^GD4689^*) wing phenotypes are highlighted. Boxplots indicate median (center line), 25th and 75th percentiles (bounds of box), and minimum and maximum (whiskers), with red dotted lines representing the control median. (D) Representative brightfield adult eye (scale bar = 100 µm) and wing (scale bar = 500µm) images with tissue-specific knockdown of CNV homologs are shown. Genotypes for the eye images are: *w^1118^;GMR-GAL4/+; UAS-Dicer2/+, w^1118^;GMR-GAL4/ UAS-Lnk^KK105731^ RNAi; UAS-Dicer2/+, w^1118^;GMR-GAL4/UAS-mEFTu1^GD16961^ RNAi; UAS-Dicer2/+*. Genotypes for the wing images are: *w^1118^/ bx^MS1096^-GAL4;+; UAS-Dicer2/+, w^1118^/bx^MS1096^-GAL4; UAS-Lnk^KK105731^ RNAi/+; UAS-Dicer2/+*, and *w^1118^/bx^MS1096^-GAL4; UAS-mEFTu1^GD16961^ RNAi/+; UAS-Dicer2/+*.

lethality with ubiquitous knockdown, including *ALDOA/Ald* and *PPP4C/Pp4-19C* within 16p11.2 and *ATXN2L/Atx2* and *TUFM/mEFTu1* within distal 16p11.2 (Fig 5A). The remaining homologs that showed lethality with ubiquitous knockdown showed at least a mild qualitative or quantitative wing phenotype.

We next compared the phenotypes observed with wing-specific knockdown of fly homologs to their corresponding eye-specific knockdowns to evaluate tissue-specific effects. The

*Drosophila* eye has been classically used to evaluate neuronal phenotypes due to genetic perturbations, as individual ommatidial units within the eye contain photoreceptor neurons that project into the optic lobe within the brain [49,50]. In fact, numerous genes associated with human neurological disorders have been modeled using the fly eye [51–53]. To quantitatively assess the phenotypic severity of cellular defects with eye-specific knockdown of fly homologs, we developed a computational tool called *Flynotyper* [54] that quantifies the degree of disorganization among the ommatidia in the adult eye. We analyzed phenotypic scores obtained from *Flynotyper* for 66 RNAi lines of 45 fly homologs, including from previously published datasets [30,31,54]. We found that 37/45 homologs (82.2%) exhibited both eye and wing-specific defects (Fig 5B, S3 Fig, S6 Data). Two homologs with significant eye phenotypes did not show any wing phenotypes, including *SPNS1/spin* within distal 16p11.2 and microcephaly-associated *SLC25A19/Tpc1* [55], while five homologs only showed wing-specific phenotypes, including *CDIPT/Pis* and *YPEL3/CG15309* within 16p11.2, *FBXO45/Fsn* and *OSTalpha/CG6836* within 3q29, and *UQCRC2/UQCR-C2* (Fig 5B, S3 Fig). In particular, *UQCRC2/UQCR-C2* showed lethality with wing-specific knockdown, suggesting potential tissue-specific effects of this gene in non-neuronal cells (Fig 5B). While most homologs contributed towards both eye and wing-specific phenotypes, we observed a wide range of severity in eye phenotypes that did not correlate with the severity of quantitative or qualitative wing phenotypes (Fig 5C). For example, *TUFM/mEFTu1* showed a severe wing phenotype but only a mild increase in eye phenotypic score, while *SH2B1/Lnk*, also within the distal 16p11.2 region, showed severe rough eye phenotypes but only a mild increase in wing size (Fig 5D). Similarly, *BCL9/lgs* also showed opposing tissue-specific effects with mild qualitative wing phenotypes and severe eye phenotypes, suggesting that the role of these homologs towards development differs across tissue types.

## CNV genes show variable expression across different tissues in flies and humans

To assess how expression levels of CNV genes vary across different tissues, we first examined the expression patterns of fly homologs in larval and adult tissues using the FlyAtlas Anatomical Microarray dataset [56]. We found that 76/77 homologs with available data were expressed in at least one larval and adult tissue (S4 Fig, S7 Data). In general, we did not observe a correlation between wing phenotype severity and expression patterns of homologs in larval or adult tissues (Fig 6A). For example, 58/77 homologs (75.3%) showed ubiquitous larval expression, including both fly homologs that showed no qualitative wing phenotypes, such as *KCTD13/CG10465* within 16p11.2 and *FBXO45/Fsn*, and those with severe wing phenotypes, such as *PPP4C/Pp4-19C* and *NCBP2/Cbp20* (Fig 6A, S4 Fig). Furthermore, 30/39 homologs (76.9%) that showed eye phenotypes also had ubiquitous larval expression, providing further support to the observation that genes causing neuronal phenotypes may also contribute to developmental phenotypes in other tissues (S6 Data). Of note, 9/77 homologs (11.7%) did not have any expression in the larval central nervous system (CNS), including *FMO5/Fmo-2*, *BDH1/CG8888* within 3q29, and *TBX6/Doc2* within 16p11.2 (Fig 6A, S4 Fig). However, we observed wing phenotypes for 8/9 of these homologs, suggesting that they may contribute to tissue-specific phenotypes outside of the nervous system. Except for the epilepsy-associated *SCN1A/para* [57], which was exclusively expressed in both the larval central nervous system (CNS) and adult brain tissues, the other tested neurodevelopmental genes were also expressed in non-neuronal tissues (Fig 6A).

We further used the GTEx Consortium dataset [58] to examine tissue-specific expression of 130 human CNV genes and 20 known neurodevelopmental genes across six tissues including brain, heart, kidney, lung, liver, and muscle. We found 121/150 genes (80.7%) that were expressed in at least one adult tissue, including 49/150 genes (32.7%) that showed ubiquitous

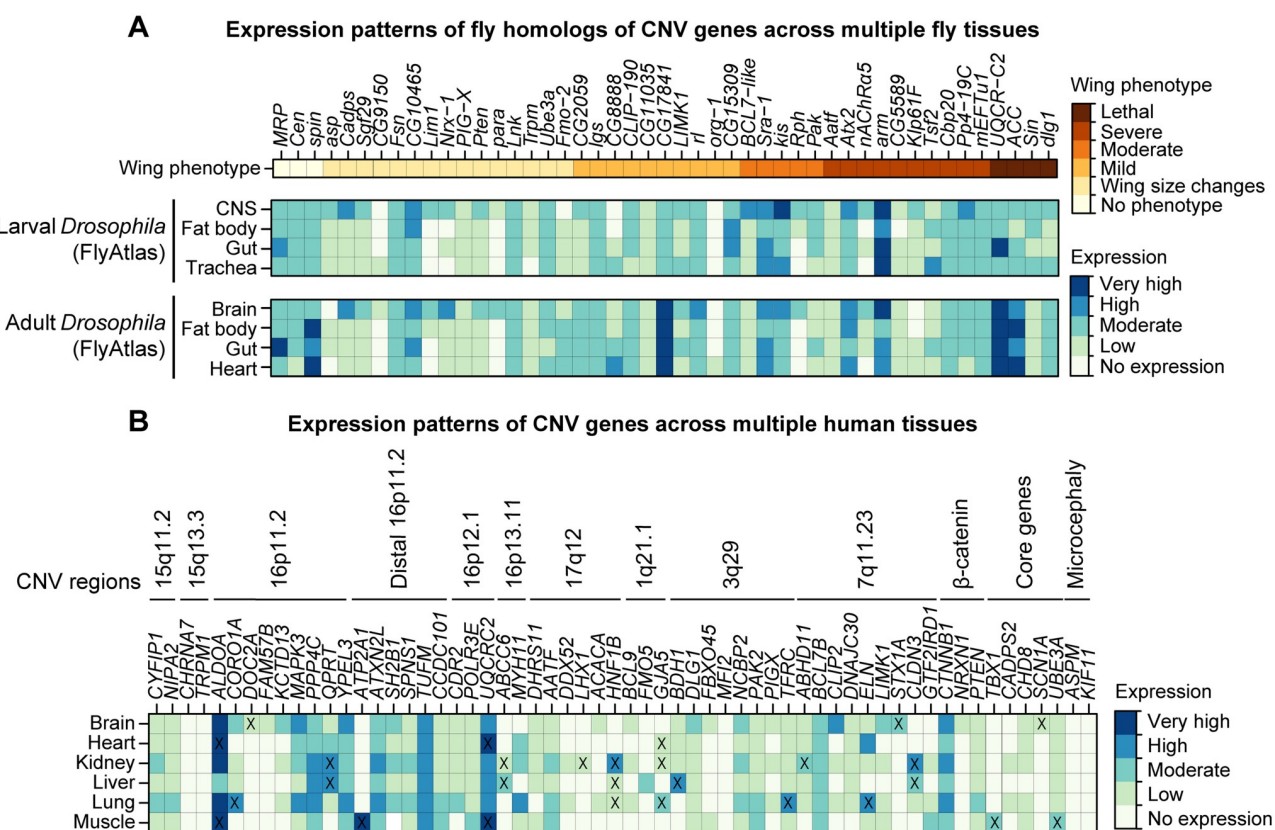

**Fig 6. Expression patterns of *Drosophila* homologs and human CNV and neurodevelopmental genes across multiple tissues.** (A) Heatmap with expression of fly homologs of select CNV and neurodevelopmental genes in multiple *Drosophila* larval and adult tissues (derived from the FlyAtlas Anatomical Microarray dataset), compared with adult wing phenotype severity, is shown. Expression values are grouped into no expression (<10 fragments per kilobase of transcript per million reads, or FPKM), low (10–100 FPKM), moderate (100–500 FPKM), high (500–1000 FPKM), and very high (>1000 FPKM) expression categories. (B) Heatmap with expression of select human CNV and neurodevelopmental genes in multiple adult tissues, derived from the Genotype-Tissue Expression (GTEx) dataset v.1.2, is shown. Expression values are grouped into no expression (<3 transcripts per million reads, or TPM), low (3–10 TPM), moderate (10–25 TPM), high (25–100 TPM), and very high (>100 TPM) expression categories. "X" symbols denote preferential expression in a particular tissue (see Methods). Expression data for all CNV and neurodevelopmental genes are provided in S7 Data.

expression across all six tissues (S6 Data). Furthermore, 112/150 genes (74.7%) were expressed in non-neuronal tissues, including 34/150 (22.7%) genes without any neuronal expression, such as *TBX1*, *FMO5* and *GJA5* within 1q21.1, and *ATP2A1* within distal 16p11.2 (Fig 6B, S7 Data). *FMO5* and *TBX1* also showed non-neuronal expression in *Drosophila* tissues, suggesting that their tissue-specific expression is highly conserved (Fig 6A). Other genes with ubiquitous expression also showed preferentially higher expression within specific non-neuronal tissues, including *ALDOA* and *UQCRC2* for muscle and heart (Fig 6B). In contrast, we found nine genes that were expressed only in the adult brain, including *FAM57B* and *DOC2A* within 16p11.2, as well as *SCN1A*, which showed similar CNS-only expression in *Drosophila* tissues (Fig 6B, S7 Data).

## Knockdown of fly homologs of CNV genes lead to disruption of cellular processes

Disruptions of basic cellular processes in neuronal cells, such as cell proliferation and apoptosis, have been implicated in neurodevelopmental disorders [59–61]. We previously identified

defects in cell proliferation among photoreceptor neurons in larval eye discs with knockdown of 16p11.2 homologs, as well as increased apoptosis with knockdown of a subset of 3q29 homologs [30,31]. Here, we explored how these basic cellular processes are altered in non-neuronal cells, specifically in the developing wing disc, with knockdown of homologs of CNV genes. We targeted 27 fly homologs that showed a range of adult wing phenotypes for changes in cell proliferation and apoptosis, using immunostaining for anti-phospho-Histone H3 Ser10 (pH3) and anti-*Drosophila* caspase-1 (dcp1), respectively, in the third instar larval wing discs. We identified 23/27 homologs that showed significant increases in apoptotic cells compared to controls, including seven homologs, such as *PPP4C/Pp4-19C*, *ATXN2L/Atx2*, and *AATF/Aatf*, which showed dcp1 staining across the entire larval wing pouch (Fig 7A and 7B, S5 and S6 Figs, S8 Data). In addition, 16/27 homologs showed decreased levels of proliferation, including eight homologs which also showed apoptosis defects, such as *CYFIP1/Sra-1* within 15q11.2, *SH2B1/Lnk*, and the microcephaly gene *KIF11/Klp61F* (Fig 7A and 7C, S5 and S6 Figs, S8 Data). All six of the tested homologs with severe adult wing phenotypes showed both increased apoptosis and decreased proliferation (S8 Data). Similarly, 3/4 homologs showing lethality with wing-specific knockdown also showed defects in apoptosis or proliferation, with the exception of *ACACA/ACC* (S5 Fig, S8 Data). As *bx^{MS1096}*-GAL4 is located on the X-chromosome, we expected to see more severe defects in males compared with females due to X-linked dosage compensation [43,44]. However, knockdown of 3/11 tested homologs with sex-specific differences in adult wing phenotypes, including *BCL9/lgs*, *CYFIP1/Sra-1*, and *DNAJC30/CG11035* within 7q11.23, showed significantly decreased levels of cell proliferation in females but no changes in males compared to their respective controls, suggesting sex-specific effects of these genes towards cell proliferation (S6 Fig, S8 Data). Overall, our results suggest that cell proliferation and apoptosis are disrupted by reduced expression of homologs of CNV genes in both neuronal and non-neuronal tissues.

## Knockdown of homologs of CNV genes disrupt conserved signaling pathways

Several conserved signaling pathways that are active in a spatial and temporal manner in the larval wing disc, such as Wnt, Hedgehog, BMP, and Notch signaling, regulate the anterior-posterior (A/P) and dorsal-ventral (D/V) boundaries to determine accurate morphology and vein patterning in the adult wing [37,38,62–64]. For example, Wnt and Notch signaling pathways act along the D/V boundary to determine cell fate [65,66], while Hedgehog signaling is dependent upon expression of both engrailed in the posterior compartment and patched along the A/P border [67,68]. Moreover, O'Roak and colleagues showed that genes carrying *de novo* mutations in patients with autism are linked to β-catenin/Wnt pathway [69]. In addition, familial loss-of-function mutations in the human Hedgehog signaling pathway gene *PTCH1* have been implicated in basal cell nevus syndrome, which leads to basal cell carcinoma [70,71].

Based on adult wing phenotypes and disruptions to cellular processes, we next tested whether knockdown of 14 fly homologs disrupt conserved signaling pathways in the third instar larval wing disc (S8 Data). In particular, we evaluated the role of Wnt, Hedgehog, and Notch signaling pathways by testing the expression patterns of four key proteins within these pathways, including wingless (Wnt), patched (Hedgehog), engrailed (Hedgehog), and delta (Notch). We found that 9/14 homologs, including 8/10 homologs showing severe wing phenotypes or lethality, exhibited disruptions in at least one signaling pathway (S8 Data). For example, five homologs with severe or lethal phenotypes showed disruptions of all four signaling pathways, including *AATF/Aatf*, *NCBP2/Cbp20*, *POLR3E/Sin*, *PPP4C/Pp4-19C*, and *KIF11/Klp61F* (Fig 8, S8 Data). Our observations are in concordance with previous findings by

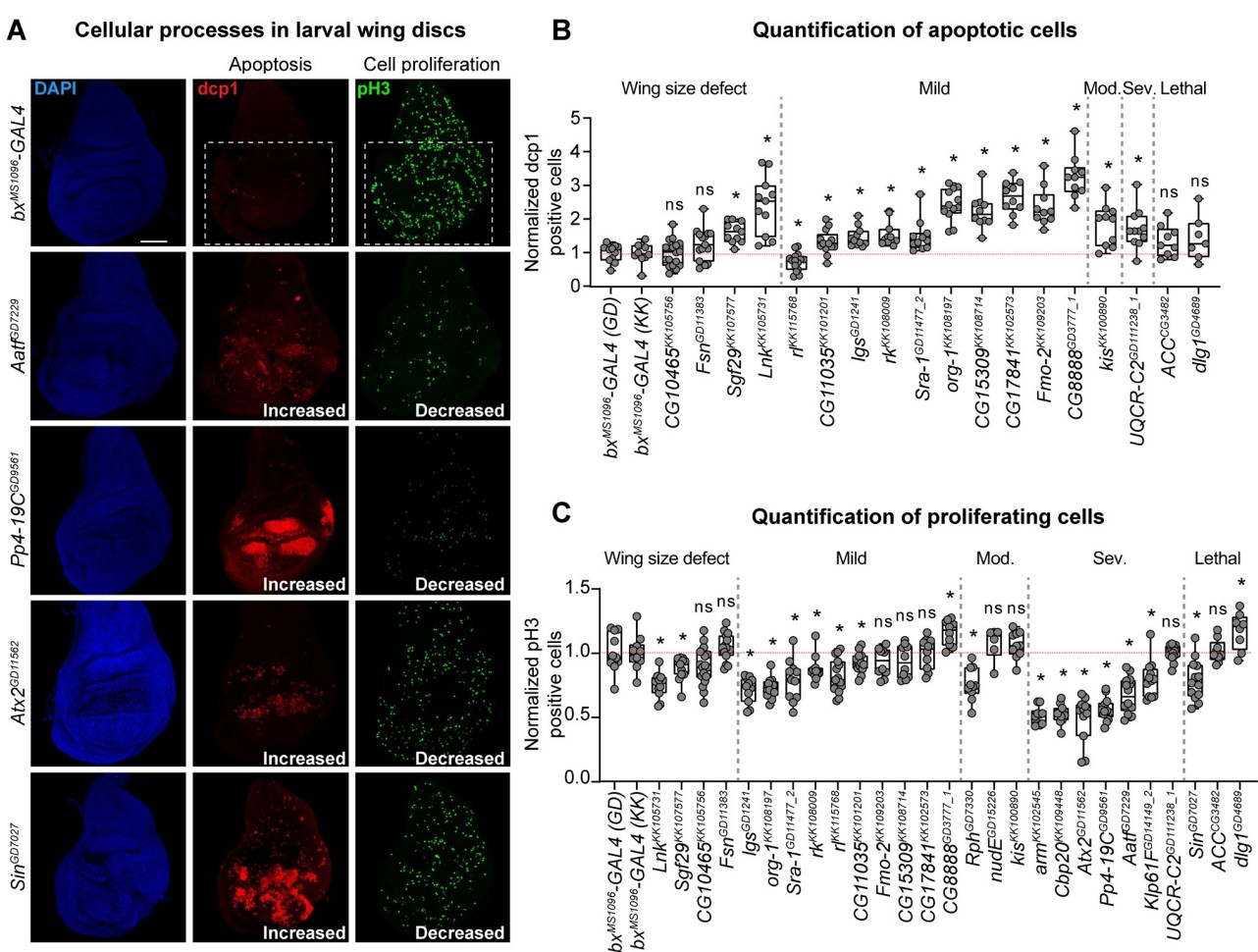

**Fig 7. *Drosophila* homologs of CNV and neurodevelopmental genes show altered levels of apoptosis and proliferation. (A)** Larval imaginal wing discs (scale bar = 50 μm) stained with nuclear marker DAPI, apoptosis marker dcp1, and cell proliferation marker pH3 illustrate altered levels of apoptosis and cell proliferation due to wing-specific knockdown of select fly homologs of CNV genes. We quantified the number of stained cells within the wing pouch of the wing disc (white box), which becomes the adult wing. Additional representative images of select homologs are presented in S5 and S6 Figs. Genotypes for the wing images are: $w^{1118}$/$bx^{MS1096}$-GAL4;+; UAS-Dicer2/+, $w^{1118}$/$bx^{MS1096}$-GAL4;+; UAS-Aatf$^{GD7229}$ RNAi/UAS-Dicer2, $w^{1118}$/$bx^{MS1096}$-GAL4;UAS-Pp4-19C$^{GD9561}$RNAi/+; UAS-Dicer2/+, $w^{1118}$/$bx^{MS1096}$-GAL4;+; UAS-Atx2$^{GD11562}$ RNAi/UAS-Dicer2, and $w^{1118}$/$bx^{MS1096}$-GAL4;+; UAS-Sin$^{GD7027}$ RNAi/UAS-Dicer2. **(B)** Boxplot of dcp1-positive cells in larval wing discs with knockdown of select fly homologs of CNV and neurodevelopmental genes, normalized to controls, is shown (n = 7–18, *p < 0.05, two-tailed Mann–Whitney test with Benjamini-Hochberg correction). We note that several RNAi lines showed severe dcp1 staining across the entire wing disc and could not be quantified. **(C)** Boxplot of pH3-positive cells in the larval wing discs with knockdown of select fly homologs of CNV and neurodevelopmental genes, normalized to controls, is shown (n = 6–18, *p < 0.05, two-tailed Mann–Whitney test with Benjamini-Hochberg correction). Boxplots are organized by adult wing phenotype (wing size defect, mild, moderate (mod.), severe (sev.), lethal). Boxplots indicate median (center line), 25th and 75th percentiles (bounds of box), and minimum and maximum (whiskers), with red dotted lines representing the control median.

Swarup and colleagues, who showed that *PPP4C/Pp4-19C* is a candidate regulator of Wnt and Notch signaling pathways in *Drosophila* larval wing discs [72]. Furthermore, two genes from the 3q29 region, *DLG1/dlg1* and *MFI2/Tsf2*, showed altered expression patterns for delta and patched but not for engrailed, indicating that they selectively interact with the Hedgehog as well as Notch signaling pathways (S7 Fig). In fact, Six and colleagues showed that Dlg1 directly binds to the PDZ-binding domain of Delta1 [73]. In contrast, *ACACA/ACC* and *UQCRC2/ UQCR-C2* showed no changes in expression patterns for any of the four signaling proteins tested, suggesting that the observed lethality could be due to other cellular mechanisms

## Disruption of signaling pathways in larval wing discs

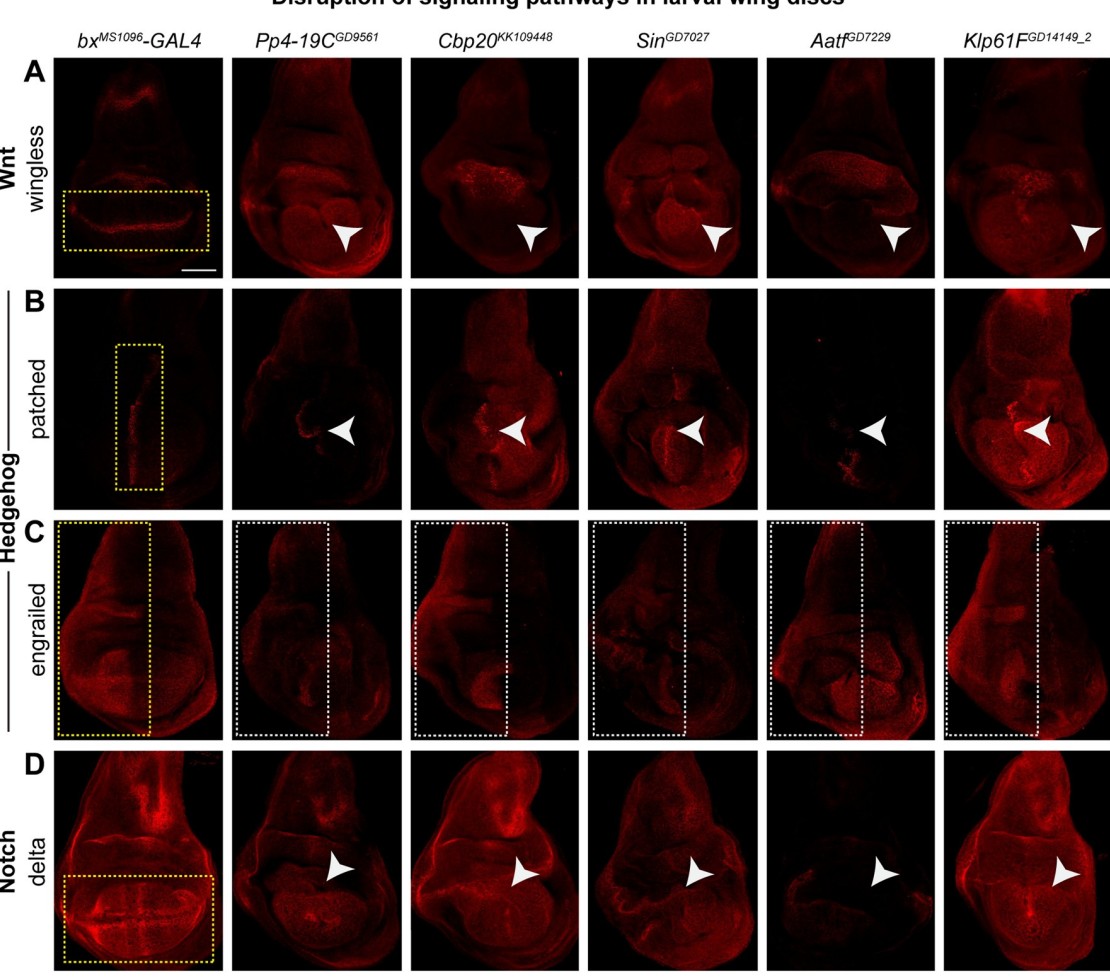

**Fig 8. Candidate *Drosophila* homologs of genes within CNV regions interact with conserved signaling pathways.** Larval imaginal wing discs (scale bar = 50 μm) stained with **(A)** wingless, **(B)** patched, **(C)** engrailed, and **(D)** delta illustrate disrupted expression patterns for proteins located within the Wnt (wingless), Hedgehog (patched and engrailed), and Notch (delta) signaling pathways due to wing-specific knockdown of select fly homologs. Dotted yellow boxes represent expression patterns for signaling proteins in $bx^{MS1096}$-GAL4 control images. White arrowheads and dotted white boxes highlight disruptions in expression patterns of signaling proteins with knockdown of homologs of CNV or neurodevelopmental genes. Additional representative images of select homologs are presented in S7 Fig. Genotypes for the wing images are: $w^{1118}/bx^{MS1096}$-GAL4;+; UAS-Dicer2/+, $w^{1118}/bx^{MS1096}$-GAL4; UAS-Pp4-19C$^{GD9561}$ RNAi/+; UAS-Dicer2/+, $w^{1118}/bx^{MS1096}$-GAL4;UAS-Cbp20$^{KK109448}$ RNAi/+; UAS-Dicer2/+, $w^{1118}/bx^{MS1096}$-GAL4;+; UAS-Sin$^{GD7027}$ RNAi/UAS-Dicer2, $w^{1118}/bx^{MS1096}$-GAL4;+; UAS-Aatf$^{GD7229}$ RNAi/UAS-Dicer2, and $w^{1118}/bx^{MS1096}$-GAL4; UAS-Klp61F$^{GD14149}$ RNAi/+; UAS-Dicer2/+.

(S7 Fig). We conclude that several tested homologs disrupt the expression of key proteins in signaling pathways in the developing larval wing discs, potentially accounting for the developmental defects observed in the adult wings.

## Connectivity patterns of CNV genes vary across human tissue-specific networks

We examined patterns of connectivity for nine candidate genes, based on disruptions of signaling pathways identified in developing *Drosophila* wing discs, within the context of human brain, heart, and kidney-specific gene interaction networks [74]. These tissue-specific

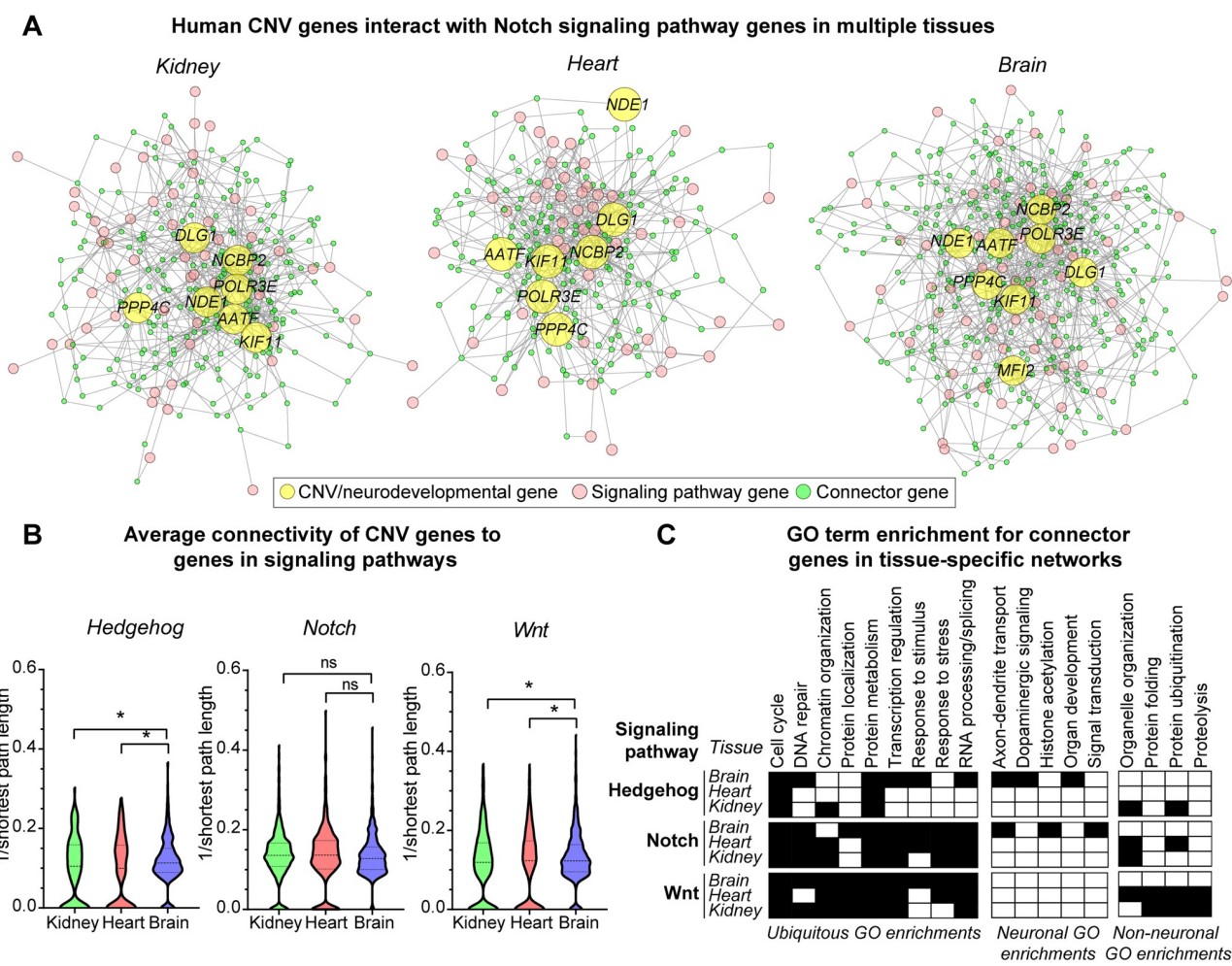

**Fig 9. Connectivity of human CNV genes with conserved signaling pathway genes in human tissue-specific networks.** (A) Network diagrams representing the connectivity of eight human CNV and neurodevelopmental genes whose fly homologs disrupt the Notch signaling pathway to 57 human Notch signaling genes within kidney, heart, and brain-specific gene interaction networks are shown. Yellow nodes represent CNV and neurodevelopmental genes, pink nodes represent Notch signaling pathway genes, and green nodes represent connector genes located within the shortest paths between CNV and Notch pathway genes. (B) Violin plots show the average connectivity (i.e. inverse of shortest path lengths) of human CNV genes to genes in the Hedgehog, Notch, and Wnt signaling pathways across three tissue-specific networks (n = 322–810 pairs of genes, *p < 0.05, two-tailed Welch's t-test with Benjamini-Hochberg correction). (C) Table shows clusters of enriched Gene Ontology (GO) Biological Process terms for connector genes observed for each signaling pathway in the three tested tissue-specific networks, categorized by enrichments in ubiquitous, neuronal, and non-neuronal tissues (p<0.05, Fisher's Exact test with Benjamini-Hochberg correction).

networks were constructed using Bayesian classifier-generated probabilities for pairwise genetic interactions based on co-expression data [74]. Within each network, we calculated the lengths of the shortest paths between each candidate gene and 267 genes from the Wnt, Notch, and Hedgehog pathways as a proxy for connectivity (S9 Data). In all three networks, each of the candidate genes were connected to a majority of the signaling pathway genes (Fig 9A, S8 Fig). Interestingly, we observed a higher connectivity (i.e. shorter path distances) between candidate genes and Wnt and Hedgehog pathway genes in the brain-specific network compared to the heart and kidney-specific networks (Fig 9B). We further identified enrichments for genes involved in specific biological processes among the connector genes that were located in the shortest paths within neuronal and non-neuronal tissue-specific networks (Fig 9C, S9 Data). For example, axon-dendrite transport, dopaminergic signaling, and signal transduction

functions were enriched among connector genes only for the brain-specific network, while organelle organization and protein ubiquitination were enriched among connector genes only for kidney and heart networks (Fig 9C). However, several core biological processes, such as cell cycle, protein metabolism, transcriptional regulation, and RNA processing/splicing, were enriched among connector genes within all three tissue-specific networks (Fig 9C). Our analysis highlights that human CNV genes potentially interact with developmental signaling pathways in a ubiquitous manner, but may affect different biological processes in neuronal and non-neuronal tissues.

## Discussion

We used the *Drosophila* wing as a model to assess how homologs of key CNV genes contribute towards non-neuronal phenotypes. We tested fly homologs of 79 genes and identified multiple homologs within each CNV region that exhibited robust phenotypes indicative of developmental disruptions. Several themes have emerged from our study highlighting the importance of fly homologs of CNV genes towards both global and tissue-specific phenotypes.

*First*, we found that fly homologs of CNV genes contribute towards developmental phenotypes through ubiquitous roles in neuronal and non-neuronal tissues. Although we did not study models for the entire CNV, nearly all individual fly homologs of CNV genes contribute to wing-specific developmental defects. It is likely that these genes also contribute to additional phenotypes in other tissues that we did not assess. In fact, a subset of these homologs also showed early lethality with ubiquitous knockdown in addition to severe or lethal wing-specific phenotypes. However, we found no correlation between the severity of the eye and wing phenotypes, suggesting tissue-specific effects of these homologs towards developmental defects. In contrast, fly homologs of known neurodevelopmental genes generally showed milder wing phenotypes compared with eye phenotypes, indicating a more neuronal role for these genes. While our study only examined a subset of CNV genes with *Drosophila* homologs, phenotypic data from knockout mouse models also support a global developmental role for individual CNV genes. In fact, 44/130 (33.8%) knockout models of CNV genes within the Mouse Genome Informatics (MGI) database [75] exhibited non-neuronal phenotypes, including 20 homologs of CNV genes that showed both neuronal and non-neuronal phenotypes (S5 Data). For example, knockout mouse models of $Dlg1^{-/-}$ showed defects in dendritic growth and branching in the developing nervous system, in addition to craniofacial features and multiple kidney and urinary tract defects [76–79]. Furthermore, Chapman and colleagues showed that knockout of $Tbx6^{-/-}$ caused defects in mesodermal and neuronal differentiation early in development, leading to abnormal vascular, tail bud, and neural tube morphology [80]. These observations further support our findings that most fly homologs of CNV genes have a global role in development that could account for the observed non-neuronal defects.

*Second*, based on tissue-specific phenotypes, we identified fly homologs of CNV genes that are key regulators of conserved cellular processes important for development. For example, 9/10 homologs with severe or lethal adult wing phenotypes also exhibited defects in cell proliferation and apoptosis during development. In fact, we found concordance between cellular processes affected by wing and eye-specific knockdown of homologs of genes within 16p11.2 and 3q29 regions, including decreased proliferation for *MAPK3/rl* and increased apoptosis for *NCBP2/Cbp20* and *DLG1/dlg1* [30,31]. While eye-specific knockdown of *BDH1/CG8888* showed decreased cell proliferation in larval eye discs [31], we found increased cell proliferation with wing-specific knockdown, suggesting a tissue-specific effect for this gene. Notably, at least one fly homolog per CNV region showed defects in cell proliferation or apoptosis, suggesting these cellular processes are important for development in both neuronal and non-

neuronal tissues. For example, *ATXN2L/Atx2*, *SH2B1/Lnk*, and *CCDC101/Sgf29* each showed decreased proliferation and increased apoptosis, suggesting a potential shared cellular mechanism for several genes within the distal 16p11.2 deletion. Furthermore, a subset of these genes also disrupted multiple signaling pathways, indicating a potential role for these homologs as key regulators of developmental processes. We specifically identified five homologs whose knockdown caused disruptions of Wnt, Notch, and Hedgehog signaling pathways. Each of these genes have important roles in cell cycle regulation, apoptosis, transcription, or RNA processing, based on Gene Ontology annotations [81,82]. In fact, we found that the RNA transport protein *NCBP2/Cbp20* [83], which we recently identified as a key modifier gene for the 3q29 deletion [31], interfaced with all three signaling pathways. Furthermore, *AATF* disrupts apoptosis and promotes cell cycle progression through displacement of HDAC1 [84–86], while *PPP4C* promotes spindle organization at the centromeres during mitosis [87]. While we only evaluated the role of these genes towards development in a single fly tissue, our additional analysis of human gene interaction networks showed strong connectivity between the CNV genes and signaling pathways in multiple neuronal and non-neuronal human tissues. In fact, cell cycle genes were enriched among the connector genes in all three tested tissue-specific networks, further emphasizing the role of cell cycle processes towards developmental phenotypes. Notably, we also observed certain biological processes enriched among connector genes that were specific to neuronal or non-neuronal tissues, indicating that genes within CNV regions may affect different biological processes in a tissue-specific manner.

*Overall*, we show that fly homologs of most CNV genes contribute towards global developmental phenotypes, although exactly how they contribute toward such phenotypes varies between neuronal and non-neuronal tissues. Previous functional studies for CNV disorders have focused primarily on identifying candidate genes for the observed neuronal phenotypes. In this study, we identified several homologs of CNV genes that are responsible for non-neuronal defects, as well as novel associations between these homologs and conserved biological processes and pathways. We therefore propose that multiple genes within each CNV region differentially disrupt conserved cellular pathways and biological processes in neuronal versus non-neuronal tissues during development (Fig 10). These results are in line with a multigenic model for CNV disorders, as opposed to models where individual causative genes are responsible for specific phenotypes [30,31,88]. Our study further exemplifies the utility of evaluating non-neuronal phenotypes in addition to neuronal phenotypes in functional models of individual genes and CNV regions associated with developmental disorders, including future studies in mammalian or cellular model systems. Further studies exploring how CNV genes interact with each other and with other developmental pathways could more fully explain the conserved mechanisms underlying global developmental defects and identify potential therapeutic targets for these disorders.

## Materials and methods

### Fly stocks and genetics

We selected *Drosophila* homologs for 59 human genes out of 130 total genes located within 10 pathogenic CNV regions [89] associated with neurodevelopmental disorders (1q21.1, 3q29, 7q11.23, 15q11.2, 15q13.3, 16p11.2, distal 16p11.2, 16p12.1, 16p13.11, and 17q12) (S2 Data). These 59 genes were selected based on the presence of fly homologs, availability of RNAi lines, and quality control of the RNAi lines. In addition, we evaluated fly homologs of 20 human genes known to be in involved in neurodevelopmental disorders [54,90] (S2 Data). These include genes involved in the β-catenin signaling pathway (5 genes), core genes implicated in neurodevelopmental disorders (8 genes), and genes associated with microcephaly (7 genes)

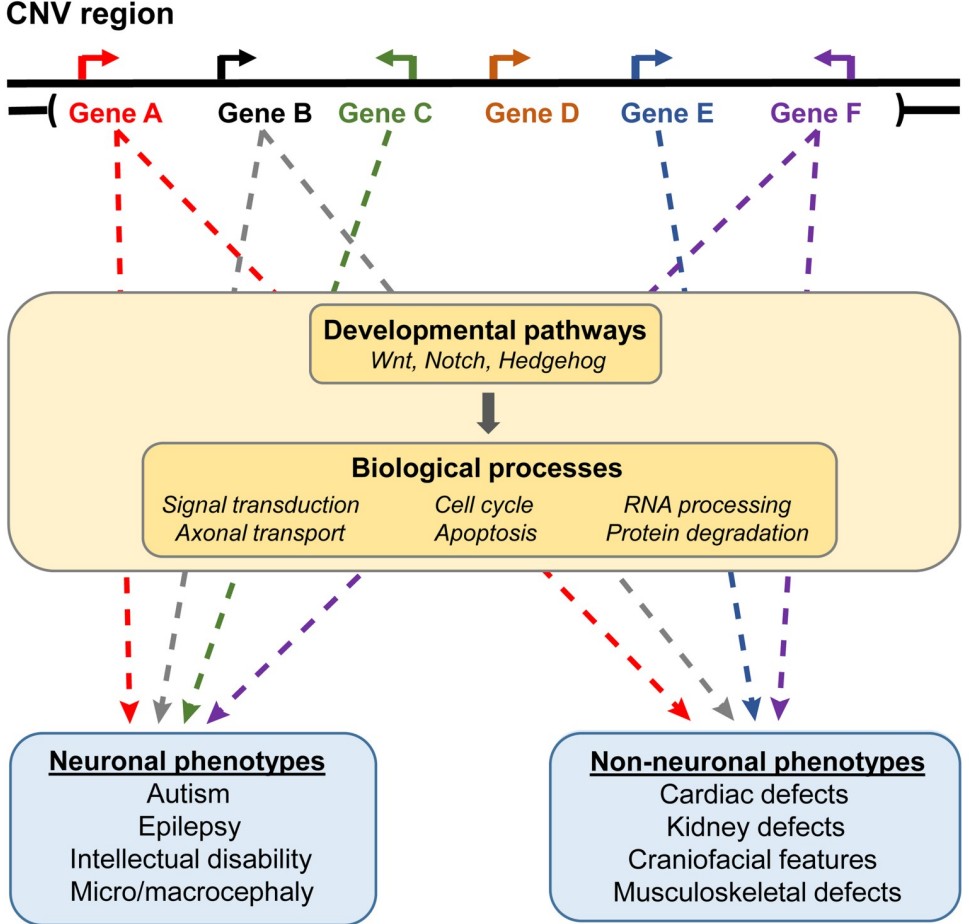

**Fig 10. Multiple genes contribute towards neuronal and non-neuronal phenotypes associated with pathogenic CNVs.** While a subset of genes within CNV regions contribute towards tissue-specific phenotypes, a majority of genes contribute towards both neuronal and non-neuronal phenotypes through disruption of developmental signaling pathways and global biological processes.

[91]. We note that 2/20 neurodevelopmental genes, *TBX1* (22q11.2 deletion) [92] and *UBE3A* (Prader-Willi/Angelman syndrome region) [93], are located within other pathogenic CNV regions. We used the DRSC Integrative Ortholog Prediction Tool (DIOPT, v.7.1) [42] to select fly homologs for each human gene based on highest numeric score, excluding genes with "low" rankings (score <2) (S2 Data).

To knockdown individual genes in specific tissues, we used RNA interference (RNAi) and the *UAS-GAL4* system (Fig 1), a well-established tool that allows for tissue-specific expression of a gene of interest [94]. RNAi lines were obtained from Vienna *Drosophila* Resource Center (VDRC), including both GD and KK lines. We tested a total of 136 lines in our data analysis (S10 Data). A complete list of stock numbers and full genotypes for all RNAi lines used in this study is presented in S10 Data. We used the *bx^MS1096^-GAL4/FM7c;;UAS-Dicer2/TM6B* driver for wing-specific knockdown and *w^1118^;GMR-GAL4;UAS-Dicer2* driver (Claire Thomas, Penn State University) for eye-specific knockdown of RNAi lines. Ubiquitous knockdown experiments were performed using the *w;da-GAL4;+* driver (Scott Selleck, Penn State University). For all experiments, we used appropriate GD (*w^1118^*, VDRC# 60000) or KK (*y,w^1118^; P{attP,y^+^,*

$w^3$ }, VDRC# 60100) lines as controls to compare against lines with RNAi knockdown of individual homologs. All fly lines were reared on standard cornmeal-yeast-dextrose *Drosophila* media at room temperature. All crosses were set and maintained at 25˚C except for the eye knockdown experiments, which were maintained at 30˚C.

## Phenotypic analysis of adult wing images

Adult progeny were isolated from crosses between RNAi lines and $bx^{MS1096}$-*GAL4* driver shortly after eclosion, and kept at 25˚C until day 2–5 (Fig 1). The progeny were then frozen at -80˚C, and moved to -20˚C prior to imaging. Approximately 20–25 progeny, both male and female, were collected for each RNAi line tested. The adult wings were plucked from frozen flies and mounted on a glass slide. The slides were covered with a coverslip and sealed using clear nail polish. Adult wing images were captured at 40X magnification using a Zeiss Discovery V20 stereoscope (Zeiss, Thornwood, NY, USA), with a ProgRes Speed XT Core 3 camera and CapturePro v.2.8.8 software (Jenoptik AG, Jena, Germany).

For each non-lethal RNAi line, we scored the adult wing images for five qualitative phenotypes, including wrinkled wing, discoloration, missing veins, ectopic veins, and bristle planar polarity defects, on a scale of 1 (no phenotype) to 5 (lethal) (Fig 3C). Lines showing severely wrinkled wings or lethality were scored as 4 (severe) or 5 (lethal) for all five phenotypes. We calculated the frequency of each phenotypic score (i.e. mild bristle polarity, moderate discoloration) across all of the wing images for each line (Fig 3A and 3B, S3 Data), and then performed k-means clustering of phenotypic frequencies for female RNAi lines to generate five clusters for overall wing phenotypes (Fig 2B, S3 Data). For quantitative analysis of wing phenotypes, we used the Fiji ImageJ software [95] to calculate the wing area using the Measure Area tool, and calculated the lengths of longitudinal veins L2, L3, L4, and L5 as well as the crossveins ACV and PCV by tracing individual veins using the Segmented Line tool (Fig 4A, S3 Data). We determined discordant homologs when multiple RNAi lines for the same homolog showed no wing phenotype versus any qualitative or quantitative wing phenotype, followed by discordance for small or large wing measurement phenotypes (S4 Data).

## Phenotypic analysis of adult eye images

We crossed RNAi lines with *GMR-GAL4* to achieve eye-specific knockdown of homologs of CNV and known neurodevelopmental genes. Adult female progeny aged 2–3 days were collected, immobilized by freezing at -80˚C, and then moved to -20˚C prior to imaging. Flies were mounted on Blu-tac (Bostik Inc, Wauwatosa, WI, USA) and imaged at 0.5X magnification using an Olympus BX53 compound microscope with LMPLan N 20X air objective using a DP73 c-mount camera (Olympus Corporation, Tokyo, Japan). CellSens Dimension software (Olympus Corporation, Tokyo, Japan) was used to capture the eye images, which were then stacked using the Zerene Stacker software (Zerene Systems LLC, Richland, WA, USA). All eye images presented in this study are maximum projections of approximately 20 consecutive optical z-sections, at a z-step size of 12.1μm. Finally, we used our computational method called *Flynotyper* (http://flynotyper.sourceforge.net) to quantify the degree of rough eye phenotypes present due to knockdown of homologs of CNV or neurodevelopmental genes [54]. *Flynotyper* scores for homologs of 16p11.2 and 3q29 genes, as well as select core neurodevelopmental genes, were derived from our previous studies [30,31,54].

## Immunohistochemistry

Wing imaginal discs from third instar larvae were dissected in 1X PBS. The tissues were fixed using 4% paraformaldehyde and blocked using 1% bovine serum albumin (BSA). The wing

discs were incubated with primary antibodies using appropriate dilutions overnight at 4˚C. We used the following primary antibodies: mouse monoclonal anti-pH3 (S10) (1:100 dilutions, Cell Signaling 9706L), rabbit polyclonal anti-cleaved *Drosophila* dcp1 (Asp216) (1:100 dilutions, Cell Signaling 9578S), mouse monoclonal anti-wingless (1:200 dilutions, DSHB, 4D4), mouse monoclonal anti-patched (1:50 dilutions, DSHB, *Drosophila* Ptc/APA1), mouse monoclonal anti-engrailed (1:50 dilutions, DSHB, 4D9), and mouse monoclonal anti-delta (1:50 dilutions, DSHB, C594.9B). Following incubation with primary antibodies, the wing discs were washed and incubated with secondary antibodies at 1:200 dilution for approximately two hours at room temperature. We used the following secondary antibodies: Alexa Fluor 647 dye goat anti-mouse (A21235, Molecular Probes by Invitrogen/Life Technologies), Alexa Fluor 568 dye goat anti-rabbit (A11036, Molecular Probes by Invitrogen/Life Technologies), and Alexa Fluor 568 dye goat anti-mouse (A11031, Molecular Probes by Invitrogen/Life Technologies). All washes and antibody dilutions were made using 0.3% PBS with Triton-X.

Third instar larvae wing imaginal discs were mounted in Prolong Gold antifade reagent with DAPI (Thermo Fisher Scientific, P36930) for imaging using an Olympus Fluoview FV1000 laser scanning confocal microscope (Olympus America, Lake Success, NY). Images were acquired using FV10-ASW 2.1 software (Olympus, Waltham, MA, USA). Composite z-stack images were analyzed using the Fiji ImageJ software [95]. To calculate the number of pH3 positive cells within the wing pouch area of the wing discs, we used the AnalyzeParticles function in ImageJ, while manual counting was used to quantify dcp1-positive cells. We note that cell proliferation and apoptosis staining for *NCBP2/Cbp20*, *DLG1/dlg1*, *BDH1/CG8888*, and *FBXO45/Fsn* were previously published[31].

## Statistical analysis

Significance for the wing area and vein length measurements, cell counts for proliferation and apoptosis, and *Flynotyper* scores were compared to appropriate GD or KK controls using one-tailed or two-tailed Mann-Whitney tests. P-values for each set of experiments were corrected for multiple testing using Benjamini-Hochberg correction. All statistical and clustering analysis was performed using R v.3.6.1 (R Center for Statistical Computing, Vienna, Austria). Details for the statistical tests performed for each dataset are provided in S11 Data.

## Expression data analysis

We obtained tissue-specific expression data for fly homologs of CNV genes from the FlyAtlas Anatomical Microarray dataset [56]. Raw FPKM (fragments per kilobase of transcript per million reads) expression values for each tissue were categorized as follows: <10, no expression; 10–100, low expression; 100–500, moderate expression; 500–1000, high expression; and >1000, very high expression (S7 Data). The median expression among midgut, hindgut, Malpighian tube, and (for adult only) crop tissues was used to represent overall gut expression. We similarly obtained human tissue-specific expression data for CNV genes from the GTEx Consortium v.1.2 RNA-Seq datasets [58]. Median TPM (transcripts per million reads) expression values for each tissue were categorized as follows: <3, no expression; 3–10, low expression; 10–25, moderate expression; 25–100, high expression; and >100, very high expression (S7 Data). The median expression among all sub-types of brain and heart tissues was used to represent brain and heart expression, while the median expression among all colon, esophagus, small intestine, and stomach tissues was used to represent digestive tract expression. Tissue-specific preferential gene expression was determined if the expression of a gene within a particular tissue was greater than the third quartile of expression values for that gene across all

tissues, plus 1.5 times the interquartile range. Venn diagrams were generated using the Venny webtool (http://bioinfogp.cnb.csic.es/tools/venny) (S4 Fig).

## Network analysis

We obtained human tissue-specific gene interaction networks for brain, heart, and kidney tissues from the GIANT network database [74] within HumanBase (https://hb.flatironinstitute.org). These networks were built using a Bayesian classifier trained on tissue-specific gene co-expression datasets, which assigned posterior probabilities for interactions between pairs of genes within a particular tissue. We downloaded the "Top edge" version of each tissue-specific network, and extracted all gene pairs with posterior probabilities >0.2 to create networks containing the top ~0.5% tissue-specific interactions. Next, we identified the shortest paths in each network between human CNV genes whose fly homologs disrupted signaling pathways in the larval wing disc and human genes within each disrupted pathway, using the inverse of the posterior probabilities as weights for each interaction in the network. Gene sets from the human Notch (KEGG:map04330), Wnt (KEGG:map04310) and Hedgehog pathways (KEGG:map04340) were curated from the Kyoto Encyclopedia of Genes and Genomes (KEGG) pathway database [96]. Using the NetworkX Python package [97], we calculated the shortest distance between each CNV gene and pathway gene, and identified connecting genes that were within each of the shortest paths for the three tissue-specific networks. We further tested for enrichment of Gene Ontology (GO) terms (GO-Slim) among the connector genes using the PantherDB Gene List Analysis tool [98]. Lists of the shortest paths and connector genes in each tissue-specific network, as well as enriched GO terms for sets of connector genes, are provided in S9 Data. Gene networks were visualized in Cytoscape v.3.7.2 [99] using an edge-weighted spring embedded layout.

## Mouse and human phenotypic data analysis

Phenotypic data for mouse models of CNV gene homologs, categorized using top-level Mammalian Phenotype Ontology terms, were obtained from the Mouse Genome Informatics (MGI) database [75] (S5 Data). Phenotypic data for human carriers of 10 pathogenic CNVs were obtained from the DECIPHER public database [9] (S1 Data). Clinical phenotypes for each CNV carrier were categorized by top-level Human Phenotype Ontology terms [100] using the Orange3 Bioinformatics software library (https://orange-bioinformatics.readthedocs.io), and the frequency of individuals carrying each top-level phenotype term was calculated for each CNV.

## Code availability

All source code for data analysis in this manuscript, including scripts for k-means clustering of fly phenotypes, network connectivity of CNV and developmental pathway genes, and extraction of top-level human phenotype terms from DECIPHER clinical phenotype data, are available on the Girirajan lab GitHub page at https://github.com/girirajanlab/CNV_wing_project.

## Supporting information

**S1 Fig. Phenotypic expression of CNV carriers across tissues.** Heatmap shows frequencies of non-neuronal developmental phenotypes observed in 1,225 human carriers of 10 pathogenic CNV deletions, curated from the DECIPHER database. CNV carriers show a variety of phenotypes that manifest across different tissues, including eye, limbs, muscle, and skeleton. (PDF)

**S2 Fig. Quantitative vein length phenotypes for select *Drosophila* homologs of CNV and neurodevelopmental genes.** Boxplots show **(A)** L2, **(B)** L4, and **(C)** L5, longitudinal veins, and **(D)** anterior crossvein (ACV) and **(E)** posterior crossvein (PCV) lengths, for knockdown of select homologs in adult fly wings (n = 9–91, $^*$p < 0.05, two-tailed Mann–Whitney test with Benjamini-Hochberg correction). Boxplots indicate median (center line), 25th and 75th percentiles (bounds of box), and minimum and maximum (whiskers), with red dotted lines representing the control median.
(PDF)

**S3 Fig. Comparisons of eye-specific and wing-specific knockdowns for select *Drosophila* homologs of CNV and neurodevelopmental genes.** Boxplots show *Flynotyper*-derived phenotypic scores for 66 tested adult eyes with eye-specific knockdown (*GMR-GAL4*) of select homologs of CNV and neurodevelopmental genes, normalized as fold-change (FC) to control values (n = 1–40, $^*$p < 0.05, one-tailed Mann–Whitney test with Benjamini-Hochberg correction). RNAi lines that do not show any observable qualitative adult wing phenotypes, including lines that show wing measurement phenotypes, are represented in **(A)**, while RNAi lines with observable mild to lethal qualitative wing phenotypes are represented in **(B)**. Boxplots indicate median (center line), 25th and 75th percentiles (bounds of box), and minimum and maximum (whiskers), with red dotted lines representing the control median.
(PDF)

**S4 Fig. Expression of *Drosophila* homologs of CNV and neurodevelopmental genes in larval and adult tissues.** Venn diagrams representing the number of 76/77 fly homologs of CNV and neurodevelopmental genes that are expressed (>10 fragments per kilobase of transcript per million reads, or FPKM) in **(A)** larval (central nervous system or CNS, gut, trachea, and fat body) and **(B)** adult tissues (brain, gut, heart and fat body) are shown.
(PDF)

**S5 Fig. Additional *Drosophila* homologs of CNV and neurodevelopmental genes show altered levels of cell proliferation and apoptosis.** Larval imaginal wing discs (scale bar = 50 μm) stained with nuclear marker DAPI, apoptosis marker dcp1, and cell proliferation marker pH3 illustrate altered levels of apoptosis and cell proliferation due to wing-specific knockdown of select fly homologs of CNV and neurodevelopmental genes. We examined changes in the number of stained cells within the wing pouch of the wing disc (white box), which becomes the adult wing. Genotypes for the wing images are: $w^{1118}/bx^{MS1096}$-GAL4;+; UAS-Dicer2/+, $w^{1118}/bx^{MS1096}$-GAL4;UAS-Cbp20$^{KK109448}$ RNAi/+; UAS-Dicer2/+, $w^{1118}/bx^{MS1096}$-GAL4;+; UAS-dlg1$^{GD4689}$ RNAi/UAS-Dicer2, $w^{1118}/bx^{MS1096}$-GAL4;UAS-CG8888$^{GD3777}$ RNAi/+; UAS-Dicer2/+, $w^{1118}/bx^{MS1096}$-GAL4;+; UAS-UQCR-C2$^{GD11238}$ RNAi/UAS-Dicer2, $w^{1118}/bx^{MS1096}$-GAL4;+; UAS-ACC$^{GD3482}$ RNAi/UAS-Dicer2, $w^{1118}/bx^{MS1096}$-GAL4;UAS-Klp61F$^{GD14149}$ RNAi/+; UAS-Dicer2/+, and $w^{1118}/bx^{MS1096}$-GAL4;UAS-Rph$^{GD7330}$ RNAi/+;UAS-Dicer2/+.
(PDF)

**S6 Fig. Select female and male *Drosophila* homologs of CNV and neurodevelopmental genes show altered levels of cell proliferation and apoptosis. (A)** Larval imaginal wing discs (scale bar = 50 μm) stained with nuclear marker DAPI, apoptosis marker dcp1, and cell proliferation marker pH3 illustrate altered levels of apoptosis and cell proliferation due to wing-specific knockdown of select fly homologs of CNV genes in females and males. We examined changes in the number of stained cells within the wing pouch of the wing disc (white box), which becomes the adult wing. Genotypes for the wing images are: $w^{1118}/bx^{MS1096}$-GAL4;+; UAS-Dicer2/+, $w^{1118}/bx^{MS1096}$-GAL4;+; UAS-lgs$^{GD1241}$ RNAi/UAS-Dicer2, $w^{1118}/bx^{MS1096}$-GAL4;+; UAS-Sra-1$^{GD11477}$ RNAi/UAS-Dicer2, and $w^{1118}/bx^{MS1096}$-GAL4;

*UAS-CG11035*$^{KK101201}$ *RNAi/+; UAS-Dicer2/+.* **(B)** Boxplot shows dcp1-positive cells in larval wing discs with knockdown of select fly homologs of CNV and neurodevelopmental genes, normalized to controls (n = 9–13, $^*$p < 0.05, two-tailed Mann–Whitney test with Benjamini-Hochberg correction). **(C)** Boxplot shows pH3-positive cells in the larval wing discs with knockdown of select fly homologs of CNV and neurodevelopmental genes, normalized to controls (n = 9–13, $^*$p < 0.05, two-tailed Mann–Whitney test with Benjamini-Hochberg correction). Boxplots indicate median (center line), 25th and 75th percentiles (bounds of box), and minimum and maximum (whiskers), with red dotted lines representing the control median. (PDF)

**S7 Fig. Additional *Drosophila* homologs of genes within CNV regions interact with conserved signaling pathways to induce developmental phenotypes.** Larval imaginal wing discs (scale bar = 50 μm) stained with **(A)** wingless, **(B)** patched, **(C)** engrailed, and **(D)** delta illustrate disrupted expression patterns for proteins located within the Wnt (wingless), Hedgehog (patched and engrailed), and Notch (delta) signaling pathways due to wing-specific knockdown of additional fly homologs of CNV and neurodevelopmental genes. Dotted yellow boxes represent expression patterns for signaling proteins in *bx*$^{MS1096}$-GAL4 control images. White arrowheads and dotted white boxes highlight disruptions in expression patterns of signaling proteins with knockdown of CNV genes. Genotypes for the wing images are: *w*$^{1118}$/*bx*$^{MS1096}$-GAL4;+; *UAS-Dicer2/+, w*$^{1118}$/*bx*$^{MS1096}$-GAL4;+; *UAS-dlg1*$^{GD4689}$ *RNAi/UAS-Dicer2, w*$^{1118}$/*bx*$^{MS1096}$-GAL4;+; *UAS-Tsf2*$^{GD2442}$ *RNAi/UAS-Dicer2, w*$^{1118}$/*bx*$^{MS1096}$-GAL4;+; *UAS-Atx2*$^{GD11562}$ *RNAi/UAS-Dicer2, w*$^{1118}$/*bx*$^{MS1096}$-GAL4;+; *UAS-UQCR-C2*$^{GD11238}$ *RNAi/ UAS-Dicer2, w*$^{1118}$/*bx*$^{MS1096}$-GAL4;+; *UAS-nudE*$^{GD15226}$ *RNAi/UAS-Dicer2*, and *w*$^{1118}$/ *bx*$^{MS1096}$-GAL4;+; *UAS-ACC*$^{GD3482}$ *RNAi/UAS-Dicer2.* (PDF)

**S8 Fig. Tissue-specific network diagrams showing connectivity of human CNV genes with conserved signaling pathway genes.** Network diagrams for connectivity between nine human CNV and neurodevelopmental genes whose fly homologs disrupt the **(A)** Wnt and **(B)** Hedgehog signaling pathways and 162 human Wnt and 46 human Hedgehog signaling genes within kidney, heart, and brain-specific gene interaction networks are shown. Yellow nodes represent CNV and neurodevelopmental genes, pink nodes represent Wnt or Hedgehog signaling pathway genes, and green nodes represent connector genes within the shortest paths between CNV and signaling pathway genes. (PDF)

**S1 Data. Non-neuronal phenotypes of individuals carrying rare pathogenic CNVs in the DECIPHER database.** (XLSX)

**S2 Data. *Drosophila* homologs of human CNV and neurodevelopmental genes, as determined using DIOPT v.7.1.** Grey shading indicates human CNV genes that were not assessed in this study. (XLSX)

**S3 Data. Qualitative and quantitative adult wing phenotypes for *Drosophila* homologs of human CNV and neurodevelopmental genes.** This file shows the raw frequencies of severity categories for five qualitative wing phenotypes and average areas and vein lengths for all 136 female and male tested RNAi lines. The file also includes k-means clusters for the female RNAi lines. (XLSX)

**S4 Data. Summary of adult wing qualitative and quantitative phenotypes for tested *Drosophila* homologs.** This file summarizes qualitative k-means clustering and longitudinal L3 vein length and wing area changes for 79 tested fly homologs. We defined discordant homologs when multiple RNAi lines for the same fly homolog showed no phenotype versus any qualitative or quantitative phenotypes, followed by discordance for small or large wing size phenotypes.
(XLSX)

**S5 Data. Phenotypes of mouse knockdown models for homologs of CNV genes.** This file lists lethality and neuronal and non-neuronal phenotypes, categorized using top-level Mammalian Phenotype Ontology terms, for knockdown models of 130 mouse homologs of CNV genes. The data were derived from the Mouse Genome Informatics (MGI) database.
(XLSX)

**S6 Data. Summary of eye-specific and wing-specific phenotypes for fly homologs.** This file summarizes eye-specific and wing-specific phenotypes by severity category for fly homologs of CNV and neurodevelopmental genes. Eye phenotype severity is defined by *Flynotyper* phenotypic scores with fold-change (FC) normalization to control as follows: no change (0–1.1 FC), mild (1.1–1.5 FC), moderate (1.5–2.0 FC), and severe (>2.0 FC). Wing phenotype severity is defined by k-means clustering for qualitative phenotypes and quantitative size changes as listed in S4 Data.
(XLSX)

**S7 Data. Tissue-specific expression of *Drosophila* homologs and human CNV and neurodevelopmental genes.** This file lists expression values across multiple fly and human tissues for 79 *Drosophila* homologs and 150 human CNV and neurodevelopmental genes. Fly expression data (fragments per kilobase of transcript per million reads, or FPKM) were derived from the FlyAtlas Anatomical Microarray dataset, and human expression data (transcripts per million reads, or TPM) were derived from the Genotype-Tissue Expression (GTEx) dataset v.1.2.
(XLSX)

**S8 Data. Summary of immunostaining of the larval imaginal wing discs.** This file summarizes changes in apoptosis (27 homologs), cell proliferation (27 homologs), and Wnt, Hedgehog, and Notch signaling pathway proteins (14 homologs), along with qualitative and quantitative adult wing phenotypes (as listed in S3 Data), for female and male fly homologs.
(XLSX)

**S9 Data. Tissue-specific network connectivity for candidate CNV genes and signaling pathway genes.** This file lists the shortest path lengths between nine candidate CNV genes and 265 genes within Wnt, Hedgehog, and Notch signaling pathways for heart, kidney, and brain-specific gene interaction networks, along with the connector genes that are within the shortest paths. Enriched Gene Ontology (GO) Biological Process, Cellular Component, and Molecular Function terms for sets of connector genes for each signaling pathway in each tissue-specific network are also provided.
(XLSX)

**S10 Data. List of *Drosophila* stocks used for experiments, including VDRC stock numbers and genotypes.**
(XLSX)

**S11 Data. Statistics for all experimental data.** This file shows all statistical information (sample size, mean/median/standard deviation of datasets, test statistics, p-values, degrees of

freedom, confidence intervals, and Benjamini-Hochberg false discovery rate corrections) for all data. Statistical information for Kruskal-Wallis tests include factors, degrees of freedom, test statistics, and post-hoc pairwise Wilcoxon tests with Benjamini-Hochberg correction. (XLSX)

## Acknowledgments

The authors thank Drs. Scott Selleck and Claire Thomas for providing fly lines for the experiments, and members of the Girirajan Lab for their helpful discussions and comments on the manuscript. This study makes use of data generated by the DECIPHER community. A full list of centers who contributed to the generation of the data is available from http://decipher. sanger.ac.uk and via email from decipher@sanger.ac.uk. Funding for the DECIPHER project was provided by the Wellcome Trust.

## Author Contributions

**Conceptualization:** Tanzeen Yusuff, Matthew Jensen, Sneha Yennawar, Santhosh Girirajan.

**Data curation:** Tanzeen Yusuff, Matthew Jensen, Sneha Yennawar, Lucilla Pizzo, Siddharth Karthikeyan, Dagny J. Gould, Avik Sarker, Erika Gedvilaite, Yurika Matsui, Janani Iyer.

**Formal analysis:** Tanzeen Yusuff, Matthew Jensen, Sneha Yennawar, Lucilla Pizzo, Siddharth Karthikeyan, Dagny J. Gould, Avik Sarker, Erika Gedvilaite, Yurika Matsui, Janani Iyer, Zhi-Chun Lai, Santhosh Girirajan.

**Funding acquisition:** Santhosh Girirajan.

**Investigation:** Tanzeen Yusuff, Matthew Jensen, Sneha Yennawar, Lucilla Pizzo, Siddharth Karthikeyan, Dagny J. Gould, Avik Sarker, Erika Gedvilaite, Yurika Matsui, Janani Iyer.

**Methodology:** Tanzeen Yusuff, Matthew Jensen, Sneha Yennawar.

**Project administration:** Tanzeen Yusuff, Matthew Jensen, Santhosh Girirajan.

**Resources:** Santhosh Girirajan.

**Supervision:** Santhosh Girirajan.

**Writing – original draft:** Tanzeen Yusuff, Matthew Jensen, Santhosh Girirajan.

**Writing – review & editing:** Tanzeen Yusuff, Matthew Jensen, Santhosh Girirajan.

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
