## [Decision Letter · Decision Letter 0]

21 Feb 2020

Dear Dr Girirajan,

Thank you very much for submitting your Research Article entitled 'Drosophila models of pathogenic copy-number variant genes show global and non-neuronal defects during development' to PLOS Genetics. Your manuscript was fully evaluated at the editorial level and by independent peer reviewers. The reviewers appreciated the attention to an important topic but identified some aspects of the manuscript that should be improved.

We therefore ask you to modify the manuscript according to the review recommendations before we can consider your manuscript for acceptance. Your revisions should address the specific points made by each reviewer.

[LINK]

Yours sincerely,

Gregory P. Copenhaver

Editor-in-Chief

PLOS Genetics

Reviewer's Responses to Questions

**Comments to the Authors:**

Reviewer #1: The manuscript entitled “Drosophila models of pathogenic copy-number variant genes show global and nonneuronal defects during development” by Yusuff and colleagues seeks to identify genes within neurodevelopmental disorder-associated copy number variants that might be responsible for phenotypes observed in patients carrying these CNVs. Although most studies have focused on the genes underlying the neurodevelopmental disorder aspects, the novelty of this study is that the authors focused on what genes might underlie the non-neuronal phenotypes. The manuscript is well written and clear, and the authors have done a substantial amount of work to identify putative developmental and/or cellular pathways perturbed when the products encoded by these genes are eliminated (this is a particularly strong section of the manuscript). The model organism they use is Drosophila with its powerful genetic toolkit, allowing the authors to screen large numbers of potential candidates using tissue-specific knockdown strategies, something that could not readily be accomplished in vertebrate systems. Although this strength is somewhat counterbalanced by the evolutionary distance between flies and humans (with a reasonable number of CNV-associated human genes either being absent or showing reduced levels of conservation in flies), the novelty of the study merits consideration for publication in PLoS Genetics. Addressing the concerns listed below would help strengthen the study and provide a more compelling argument for the use of flies to validate the genes.

1. While the authors provide intriguing data that identifies some of the likely contributors to the non-neuronal phenotypes, it would strengthen the paper if the authors could include in the study knockdown of a random selection of genes from a genomic interval of similar size to the pathogenic CNVs considered. For example, if 10/12 genes of a pathogenic CNV show non-neuronal phenotypes of varying severity, would a random selection of 10/12 genes from a similarly sized different region of the genome not associated with a pathogenic CNV give a similar result, or would the number of genes associated with phenotypes be reduced in number or severity? If the latter were the case, this would provide a strong argument that the pathogenic CNVs considered in this study are indeed enriched for developmentally important genes, and that flies are a reasonable proxy for identifying them.

2. It would help to have a figure that lays out how many genes per CNV would need to be considered for knockdown (the total number), and how many of those have fly orthologs that could be tested (Table S1 could be expanded for this). In addition, any genes within the CNVs known to be involved in any of the phenotypic anomalies present in cases carrying that CNV should be highlighted. Collectively, this would help to determine whether known genes responsible for at least a subset of the phenotypes (and are thus critically important) are able to be tested in flies. Finally, along the lines of the lgs/Fmo-2 and asp, it would be nice to show that additional phenotypes of the knockdowns of such fly genes have some relationship to the phenotypic anomalies in cases (e.g., a gene knockdown in flies resulting in reduced proliferation could be connected to a hypoplastic organ in cases).

3. It is unclear why wing-specific drivers would result in lethality unless the driver is not specific for wing expression. A second driver could be used for at least a subset of genes to address this issue, and confirm/refute the initial results. If a truly wing-specific driver were used, then knockdown of these genes should reveal a phenotype (since wing development is not required for viability).

4. How well does DIOPT predict orthologues? For example, many of the genes tested are ranked as moderate, with a range of DIOPT scores. This should be explained, including how much actual conservation exists for “moderate” genes with the various DIOPT scores.

5. The eye-specific knockdown experiments are meant to look at “neuronal defects,” but it is never explained why knockdown in the eye should give rise to neuronal defects. This needs to be explained.

6. Since flies do not have structurally conserved major vertebrate organs such as kidneys and lungs, it is more challenging to test pathways involved in their morphogenesis. (minor criticism)

Reviewer #2: In this manuscript the authors investigated genes associated with CNVs that have been shown to have neuronal effects in humans. They used RNAi to analyze and catalog non-neuronal effects for 79 drosophila homologs. They investigated phenotypic effects in wings and eyes using multiple RNAi drosophila lines. The classified phenotypic effects were systematically quantified into bins of severity. Homologs were then surveyed for disruptions in various cellular functions such as proliferation and apoptosis. I think this paper was well written, did a nice job in stating the conclusions that can be made from the analysis, but I have a few comments. The vast majority of them being cosmetic.

1. Some of the numbers, I am having a hard time to keep clear. For instance, line 145-146 “We observed four clusters of RNAi lines: 75 lines with no observable qualitative phenotypes (55.2%).” That would suggest that 61 lines had observable qualitative phenotypes. but then in line 163- “For example, discoloration (87 lines in males compared to 56 lines in females)”. That suggest that you observed discoloration in 87 lines. Does clustering group the mildest qualitative phenotypes as “no observable phenotypes” bin? If so, this could be clearer.

2. Line 159-161: “We note that 18/79 fly homologs showed discordant phenotypes between two or more RNAi lines for the same gene, which could be due to differences in expression of the RNAi construct among these lines.” Can this be confirmed with qPCR to see if certain RNAi lines are less effective than others?

3. It is not clear how the final total of 59 genes were determined. Did you identify the genes that would most likely affect nonneuronal phenotypes? Were they the only ones that had homologs and RNAi lines for? If the former, this could imply that their results might be an over estimation of the proportion of CNV genes that affect development. The authors do a good job not overstepping their results, but I feel like this should be discussed or clarified.

4. You removed the KK lines with the offsite tio target. Was this confirmed using the PCR assay or the crossing 40DUAS? This could be clearer. And if you used the PCR assay, I suggest the gel be a supplementary figure.

5. Homologs had varied phenotypic effects. Looking at figure 2A, homologs associated with CNV 16p11.2 had a higher proportion of lethal and or severe phenotypes than the others. Can you compare the CNVs against each other? Does that CNV have more severe phenotypic impacts than the other CNVs. Is it possible to associate your finding with severity of the particular CNVs in humans?

Minor comments:

6. Some of the figure legends are long (like figure 1 and 4). In the case of figure 4, not much can be done but maybe for figure 1, part A could be its own figure.

7. Figure 6B and 6C could be rearranged to show the pattern described in the text. Instead of having the X axis ordered by their effects with respect to apoptosis and proliferation. Order the genes by their wing severity. As done in figure 4B.

8. In the introduction (line 119) you state you observed 79 genes. I think it should be done similar to figure 1 legend "We evaluated 59 Drosophila homologs of genes within 10 CNV regions and 20 known neurodevelopmental genes (79 total homologs)." This makes it clear what genes your looking at. Sometimes referring to 79 genes and then sometimes referring it as 59 genes +20 others made it unclear to follow the number of genes that was observed. Having it clear in the intro would help.

9. Also it is not clearly stated that the 20 neurodevelopment genes are CNV genes or not.

**Have all data underlying the figures and results presented in the manuscript been provided?**

Reviewer #1: Yes

Reviewer #2: Yes

PLOS authors have the option to publish the peer review history of their article (what does this mean?). If published, this will include your full peer review and any attached files.

Reviewer #1: No

Reviewer #2: No

---

## [Decision Letter · Decision Letter 1]

23 Apr 2020

Dear Dr Girirajan,

We are pleased to inform you that your manuscript entitled "Drosophila models of pathogenic copy-number variant genes show global and non-neuronal defects during development" has been editorially accepted for publication in PLOS Genetics. Congratulations!

Yours sincerely,

Gregory P. Copenhaver, Ph.D.

Editor-in-Chief

PLOS Genetics

Gregory Barsh

Editor-in-Chief

PLOS Genetics

Comments from the reviewers (if applicable):

Reviewer's Responses to Questions

**Comments to the Authors:**

Reviewer #1: In the revised manuscript entitled “Drosophila models of pathogenic copy-number variant genes show global and nonneuronal defects during development,” Yusuff and colleagues have done an exceptional job in addressing reviewer concerns. Although they were unable to find an overall enrichment of genes within pathogenic CNVs that produce severe phenotypes compared to similarlyc sized genomic intervals, they correctly point out that a significant percentage of genes within the fly genome are important developmental regulators suggesting that their distribution would be located throughout the genome. They also note that CNV generation is driven by the particular local genomic structure (seg dups, repeat sequences) implying that more pathogenic CNVs would likely be found in additional regions of the genome if the appropriate underlying genomic architecture was in place. Summarizing the data, Figure R1 is a useful addition to the manuscript.

Second, the reviewers have added the requested level of detail to Table R1 to better articulate which genes within a CNV were tested, which were not, the percentage identity of the orthologues, etc. Moreover, a better description of DIOPT has been provided, as well as a description of the neuronal components of the fly eye.

While I would have liked to see a second wing-specific driver used (given the lethal phenotypes associated with the bx MS1096) in order to sort out leaky non-wing expression issues, the overall effort of the authors to address the other concerns outweighs this relatively minor issue.

**Have all data underlying the figures and results presented in the manuscript been provided?**

Reviewer #1: Yes

PLOS authors have the option to publish the peer review history of their article (what does this mean?). If published, this will include your full peer review and any attached files.

Reviewer #1: No

**Data Deposition**

http://datadryad.org/submit?journalID=pgenetics&manu=PGENETICS-D-19-02055R1

**Press Queries**

---

## [Editor Report · Acceptance letter]

5 Jun 2020

PGENETICS-D-19-02055R1 

Drosophila models of pathogenic copy-number variant genes show global and non-neuronal defects during development 

Dear Dr Girirajan, 

We are pleased to inform you that your manuscript entitled "Drosophila models of pathogenic copy-number variant genes show global and non-neuronal defects during development" has been formally accepted for publication in PLOS Genetics! Your manuscript is now with our production department and you will be notified of the publication date in due course.

With kind regards,

Kaitlin Butler

PLOS Genetics

On behalf of:
